# Log-Linear Attention

**Han Guo**[1*]  **Songlin Yang**[1*]  **Tarushii Goel**[1]  **Eric P. Xing**[3]  **Tri Dao**[2]  **Yoon Kim**[1]

[1]Massachusetts Institute of Technology  [2]Princeton University, Together AI

[3]Carnegie Mellon University, Mohamed bin Zayed University of AI, GenBio AI

`hanguo@mit.edu`

## Abstract

The attention mechanism in Transformers is an important primitive for accurate and scalable sequence modeling. Its quadratic-compute and linear-memory complexity however remain significant bottlenecks. Linear attention and state-space models enable linear-time, constant-memory sequence modeling and can moreover be trained efficiently through matmul-rich parallelization across sequence length. However, at their core these models are still RNNs, and thus their use of a fixed-size hidden state to model the context is a fundamental limitation. This paper develops log-linear attention, an attention mechanism that balances linear attention's efficiency and the expressiveness of softmax attention. Log-linear attention replaces the fixed-size hidden state with a logarithmically growing set of hidden states. We show that with a particular growth function, log-linear attention admits a similarly matmul-rich parallel form whose compute cost is log-linear in sequence length. Log-linear attention is a general framework and can be applied on top of existing linear attention variants. As case studies, we instantiate log-linear variants of two recent architectures—Mamba-2 and Gated DeltaNet—and find they perform well compared to their linear-time variants.[1]

## 1 Introduction

The attention layer (Bahdanau et al., 2014) is a core building block of modern deep learning architectures, most notably in the Transformer architecture (Vaswani et al., 2017). For training, attention can be parallelized across sequence length through reformulating the computation as a series of matrix-matrix multiplications (matmuls), which can enable efficient training on modern accelerators such as GPUs and TPUs. However, the compute cost of attention grows quadratically and its memory cost grows linearly with respect to sequence length; despite the wallclock efficiency improvements obtained from hardware-optimized implementations (Dao et al., 2022b; Dao, 2024; Shah et al., 2024; Liu et al., 2024; Kwon et al., 2023), this quadratic-compute linear-memory cost is a fundamental limitation in enabling new applications and serves as a significant bottleneck in existing ones.

Linear attention (Katharopoulos et al., 2020) replaces the softmax kernel with a simple linear kernel (i.e., dot product) to derive the "attention" scores. The use of a linear kernel makes it possible to reformulate linear attention as a linear RNN with matrix-valued hidden states, and thus linear attention enables linear-time, constant-memory sequence modeling.[2] For training, linear attention can be parallelized across sequence length via a chunking mechanism where a sequence is split up into chunks and the computations across chunks are performed in parallel (Hua et al., 2022; Sun et al., 2023; Yang et al., 2024b; Dao & Gu, 2024). The complexity of this chunkwise parallel algorithm is subquadratic in sequence length but still rich in matmuls,[3] leading to hardware-efficient implementations (Yang & Zhang, 2024; Qin et al., 2024a; Beck et al., 2025a) that obtain practical wallclock improvements over optimized implementations of softmax attention. While early versions of linear attention generally underperformed softmax attention (Kasai et al., 2021; Peng et al., 2021;

---

[*]Equal contribution.

[1]Code available at `https://github.com/HanGuo97/log-linear-attention`.

[2]Thus there are three senses in which linear attention is *linear*: the use of a linear kernel, its reformulation as a linear RNN where the hidden state is a linear function of the previous state, and its linear-time complexity.

[3]Unlike parallel scan (Blelloch, 1990) which can also parallelize linear attention across sequence length but consists mostly of elementwise operations instead of matmuls.

| Model | A | M (Data Dependent?) | Training Algorithm / Time | Decoding Time and Space | |
|---|---|---|---|---|---|
| Attention | $\sigma(\mathbf{QK}^\top)$ | Mask (✗) | FlashAttention $\mathcal{O}(T^2)$ | $\mathcal{O}(T)$ | $\mathcal{O}(T)$ |
| Linear Attention | $\mathbf{QK}^\top$ | Mask (✗) | Chunk-recurrent $\mathcal{O}(T)$ | $\mathcal{O}(1)$ | $\mathcal{O}(1)$ |
| RetNet | $\mathbf{QK}^\top$ | Semiseparable (✗) | Chunk-recurrent $\mathcal{O}(T)$ | $\mathcal{O}(1)$ | $\mathcal{O}(1)$ |
| Mamba-2 | $\mathbf{QK}^\top$ | Semiseparable (✓) | Chunk-recurrent $\mathcal{O}(T)$ | $\mathcal{O}(1)$ | $\mathcal{O}(1)$ |
| Multi-Hyena | $\mathbf{QK}^\top$ | Toeplitz (✗) | FFT $\mathcal{O}(T \log T)$ | $\mathcal{O}(\log^2 T)$ | $\mathcal{O}(T)$ |
| DeltaNet | $\mathcal{T}_{\mathbf{K}}(\mathbf{QK}^\top)$ | Mask (✗) | Chunk-recurrent $\mathcal{O}(T)$ | $\mathcal{O}(1)$ | $\mathcal{O}(1)$ |
| Gated DeltaNet | $\mathcal{T}_{\mathbf{K}}(\mathbf{QK}^\top)$ | Semiseparable (✓) | Chunk-recurrent $\mathcal{O}(T)$ | $\mathcal{O}(1)$ | $\mathcal{O}(1)$ |
| Log-Linear Mamba-2 | $\mathbf{QK}^\top$ | Hierarchical (✓) | Chunk-scan $\mathcal{O}(T \log T)$ | $\mathcal{O}(\log T)$ | $\mathcal{O}(\log T)$ |
| Log-Linear Gated DeltaNet | $\mathcal{T}_{\mathbf{K}}(\mathbf{QK}^\top)$ | Hierarchical (✓) | Chunk-scan $\mathcal{O}(T \log T)$ | $\mathcal{O}(\log T)$ | $\mathcal{O}(\log T)$ |

**Table 1:** Summary of efficient attention mechanisms under the unified formulation: $\mathbf{P} = \mathbf{A} \odot \mathbf{M}, \mathbf{O} = \mathbf{PV}$. $\mathbf{M}$ is a lower-triangle (causal) matrix. We use symbol $\mathcal{T}_{\mathbf{K}}(\mathbf{A}) = (\mathbf{A} \odot \mathbf{L})(\mathbf{I} + \mathbf{KK}^\top \odot (\mathbf{I} - \mathbf{L}))^{-1}$ for notational brevity, where $\mathbf{L}$ is a lower-triangular matrix of 1s. Here decoding time is the time per step, and decoding space refers to the overall memory complexity during generation.

Mao; Qin et al., 2022; Sun et al., 2023), modern variants with data-dependent multiplicative gates (Yang et al., 2024b; Qin et al., 2024b; Peng et al., 2024)—which include state-space models (SSMs) such as Mamba (Gu & Dao, 2024; Dao & Gu, 2024)—and delta-rule-based structured transition matrices (Schlag et al., 2021; Yang et al., 2024b;a; Grazzi et al., 2025; Siems et al., 2025; Peng et al., 2025) have led to significant improvements. However, despite these improvements linear attention's use of a fixed-sized hidden state is a fundamental limitations when it comes to certain capabilities such as associative recall over a given context (Arora et al., 2024).

This paper develops log-linear attention as a middle ground between linear attention and full softmax attention. Instead of using a single hidden state matrix to represent the history (as in linear attention/SSMs), log-linear attention maintains a growing set of hidden states where the set size grows logarithmically with respect to sequence length. With a particular choice of the growth function, we show that log-linear attention admits a matmul-rich "parallel form" for training which involves replacing the lower-triangular causal mask in ordinary linear attention with a data-dependent hierarchical matrix, which enables subquadratic training; in particular we show that the compute cost of log-linear attention is log-linear in sequence length (hence the name), while its memory cost is logarithmic. Log-linear attention is a general framework for sequence modeling and can be used to generalize existing linear attention models. As case studies, we use the framework on two popular recent models, Mamba-2 (Dao & Gu, 2024) and Gated DeltaNet (Yang et al., 2024a), to derive log-linear variants of both models, and find that these variants perform well compared to their original linear variants.

## 2 BACKGROUND: A STRUCTURED MATRIX VIEW OF EFFICIENT ATTENTION

Given an input sequence of length $T$ and the corresponding key, query, value matrices $\mathbf{K}, \mathbf{Q}, \mathbf{V} \in \mathbb{R}^{T \times d}$, softmax attention obtains the output $\mathbf{O} \in \mathbb{R}^{T \times d}$ for all time steps via $\mathbf{O} = \text{softmax}(\mathbf{QK}^\top \odot \mathbf{M})\mathbf{V}$, where $\mathbf{M} \in \{-\infty, 0\}^{T \times T}$ is a causal masking matrix. This incurs $\mathcal{O}(T^2)$ compute and $\mathcal{O}(T)$ memory, which makes it costly to apply to long sequences. As a response, there has been much recent work on efficient alternatives with sub-quadratic compute and sub-linear memory, including linear attention, state-space models, and long convolution models. Despite their differences, many of these approaches can be captured by the following equation:

$$\mathbf{P} = \mathbf{A} \odot \mathbf{M}, \quad \mathbf{O} = \mathbf{PV}, \tag{1}$$

where $\mathbf{A} \in \mathbb{R}^{T \times T}$ is an attention-like matrix (e.g., $\mathbf{QK}^\top$ in the case of ordinary linear attention) and $\mathbf{M} \in \mathbb{R}^{T \times T}$ is a lower-triangular causal masking matrix (e.g., $\mathbf{M} \in \{0, 1\}^{T \times T}$ for linear attention). By separating out the interaction terms $\mathbf{A}$ and the (potentially data-dependent) masking matrix $\mathbf{M}$, this abstraction reveals commonalities across a broad class of models, as shown in Table 1. Different structures imposed on $\mathbf{M}$ can lead to efficient training and inference algorithms. We now describe key models that fit within this framework.

**Linear attention.** Linear attention Katharopoulos et al. (2020) simply removes the softmax operation, resulting in the following parallel form[4]

$$\mathbf{O} = (\mathbf{QK}^\top \odot \mathbf{M})\mathbf{V}, \quad \mathbf{M}_{ij} = \mathbf{1}\{i \leq j\}.$$

---

[4]Here we work linear attention without any feature maps or normalization, since most recent works have found them to be unnecessary (although see (Kacham et al., 2023; Buckman et al.; Arora et al., 2024)).

Linear attention can be reparameterized into the following "recurrent form" for inference,

$$\mathbf{S}_t = \mathbf{S}_{t-1} + \boldsymbol{v}_t \boldsymbol{k}_t^\top, \quad \boldsymbol{o}_t = \mathbf{S}_t \boldsymbol{q}_t,$$

which enables linear-time constant-memory sequence modeling.

**Linear attention with (data-dependent) gates.** Vanilla linear attention lacks a forgetting mechanism, which has been shown to be crucial in ordinary RNNs. One way to incorporate such a mechanism is through a scalar gate $\alpha_t \in (0, 1)$, which results in recurrence $\mathbf{S}_t = \alpha_t \mathbf{S}_{t-1} + \boldsymbol{v}_t \boldsymbol{k}_t^\top$. This has the following corresponding parallel form:

$$\mathbf{O} = (\mathbf{Q}\mathbf{K}^\top \odot \mathbf{M})\mathbf{V}, \quad \mathbf{M}_{ij} = \prod_{k=j+1}^{i} \alpha_k. \tag{2}$$

Originally introduced by Peng et al. (2021), gated linear attention has enjoyed a resurgence in recent years (Qin et al., 2024b; Peng et al., 2024; Yang et al., 2023; Katsch, 2023) and are an instance of time-varying SSMs (Gu & Dao, 2024; Dao & Gu, 2024). Well-known models in this family include RetNet (Sun et al., 2023), which uses a data-*in*dependent gate $\alpha_t = \alpha$, and Mamba-2 (Dao & Gu, 2024), which uses the above data-dependent gate. Dao & Gu (2024) show that with a scalar gating factor, $\mathbf{M}$ has a 1-semiseparable structure where every submatrix in the lower triangular portion has rank at most 1, which can enable efficient training.

**Linear attention with the delta rule.** DeltaNet (Schlag et al., 2021) is a type of linear attention layer which updates the hidden state via the delta rule (Widrow et al., 1960),[5] where the recurrent form is given by[6]

$$\mathbf{S}_t = \mathbf{S}_{t-1} \left( \mathbf{I} - \boldsymbol{k}_t \boldsymbol{k}_t^\top \right) + \boldsymbol{v}_t \boldsymbol{k}_t^\top, \quad \boldsymbol{o}_t = \mathbf{S}_t \boldsymbol{q}_t.$$

While the original work used a purely recurrent form, Yang et al. (2024b) recently show that it is possible to parallelize DeltaNet across sequence length through leveraging a compact representation of Householder matrices (Bischof & Loan, 1985; Joffrain et al., 2006), resulting in the following parallel form (cf. (Yang et al., 2024b, §3.2)):

$$\mathbf{O} = \left( \underbrace{\left(\mathbf{Q}\mathbf{K}^\top \odot \mathbf{L}\right) \left(\mathbf{I} + \mathbf{K}\mathbf{K}^\top \odot (\mathbf{L} - \mathbf{I})\right)^{-1}}_{\mathbf{A}} \odot \mathbf{M} \right) \mathbf{V}$$

where $\mathbf{L}$ and $\mathbf{M}$ are lower-triangular matrices consisting of 1s. Since $\mathbf{A}$ itself is already lower-triangular, the causal masking matrix $\mathbf{M}$ is not strictly necessary in the above. However, by changing $\mathbf{M}$ to have 1-semiseparable structure as in Mamba-2, we can recover Gated DeltaNet (Yang et al., 2024a), whose recurrence is given by $\mathbf{S}_t = \alpha_t \mathbf{S}_{t-1}(\mathbf{I} - \boldsymbol{k}_t \boldsymbol{k}_t^\top) + \boldsymbol{v}_t \boldsymbol{k}_t^\top$. Linear attention with such data-dependent structured transition matrices has been shown to be theoretically more expressive than linear attention with multiplicative gates when it comes to certain types of *state-tracking* tasks (Merrill et al., 2024; Grazzi et al., 2025; Siems et al., 2025; Peng et al., 2025), which make these layers attractive targets to generalize via our log-linear attention framework.

**Long convolution models.** Long-convolution sequence models, where the convolution kernel size equals the sequence length, can also be cast into this framework. For example, Toeplitz neural network (Qin et al., 2023) and MultiHyena Massaroli et al. (2023) layers are given by $\mathbf{O} = \left(\mathbf{Q}\mathbf{K}^\top \odot \mathbf{T}_h\right)\mathbf{V}$, where $\mathbf{T}_h$ is a causal Toeplitz matrix generated by a long convolution kernel $\boldsymbol{h} \in \mathbb{R}^T$, i.e., $\mathbf{T}_h[i, j] = \boldsymbol{h}[i - j]$ for $i \geq j$ and 0 otherwise. Other long convolutional variants like H3 (Fu et al., 2023) and Hyena (Poli et al., 2023) also admit a precise attention-style formulation (Massaroli et al., 2023). While the decoding speed of long convolution models can be improved from $\mathcal{O}(T)$ to $\mathcal{O}(\log^2 T)$ per step (Oncescu et al., 2025), their memory cost remains linear, i.e., the same as in softmax attention. However, some long convolution models such as S4 (Gu et al., 2022) admit a reparameterization into a time-invariant SSM and thus enjoy constant-memory inference. There has also been efforts to distill long convolution models into RNNs (Massaroli et al., 2023; Qin & Zhong, 2023), but these inherit the memory bottleneck of RNNs.

---

[5]Linear attention with the delta rule is also an instance of a fast-weight programmer (Schmidhuber, 1992).

[6]The actual DeltaNet recurrence is given by $\mathbf{S}_t = \mathbf{S}_{t-1}(\mathbf{I} - \beta_t \boldsymbol{k}_t \boldsymbol{k}_t^\top) + \boldsymbol{v}_t \boldsymbol{k}_t^\top$ where $\beta_t$ is a data-dependent scalar value in either $(0, 1)$ or $(0, 2)$, but we set $\beta_t = 1$ here for notational brevity.

**Relationship between masking structure and efficient algorithms.** Using an unstructured $\mathbf{M}$ (e.g., a random lower-triangular matrix) degrades both compute and memory complexity to softmax attention-levels, despite the absence of softmax; i.e., the *structure* of $\mathbf{M}$ is essential for training/inference efficiency, not just the removal of softmax. In linear attention where $\mathbf{M}$ is a lower-triangular matrix of 1's, we can compute $\mathbf{O}$ chunkwise, leading to an $\mathcal{O}(T)$ algorithm.[7] This algorithm generalizes to the gated case where $\mathbf{M}$ has 1-semiseparable structure as shown in (Dao & Gu, 2024). Long convolution models can use FFT to bring down the cost to $\mathcal{O}(T \log T)$.

## 3 LOG-LINEAR ATTENTION

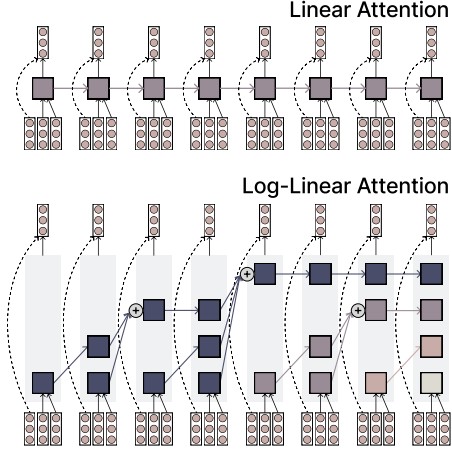

The previous section showed that the structure of the masking matrix $\mathbf{M}$ determines how compute and memory scale with sequence length. Semiseparable structures cover many efficient architectures, yielding $\mathcal{O}(T)$ training time and $\mathcal{O}(1)$ decoding memory. This motivates two questions: *(i)* what additional structures allow greater flexibility while retaining subquadratic training complexity, and *(ii)* can such models admit a recurrent form with *sublinear* decoding memory?

We answer both by introducing *log-linear attention*, which shapes $\mathbf{M}$ to achieve $\mathcal{O}(T \log T)$ computation and $\mathcal{O}(\log T)$ memory. Concretely, log-linear attention replaces the semiseparable mask with a *hierarchical* one, extending linear attention beyond semiseparable temporal structure and accommodating a broader class structures for $\mathbf{A}$. As case studies, we instantiate log-linear variants of Mamba-2 and Gated DeltaNet.

**Figure 1:** Standard linear attention (top) vs. log-linear attention (bottom). The input consists of query, key, and value vectors.

During decoding, log-linear attention employs a Fenwick tree scheme (Fenwick, 1994) that partitions inputs into power-of-two segments. Each position summarizes its prefix, enabling queries to attend to $\mathcal{O}(\log T)$ hidden states across multiple scales (Fig. 1). This design preserves fine-grained access to recent tokens while requiring only $\mathcal{O}(\log T)$ time and memory. We first focus on the simplest form of linear attention (without gating) in § 3.1 and show how log-linear attention extends it by maintaining independent recurrent states across temporal segments. Practical gated variants are presented in § 3.4.

### 3.1 FENWICK TREE PARTITIONING AND HIERARCHICAL MATRICES

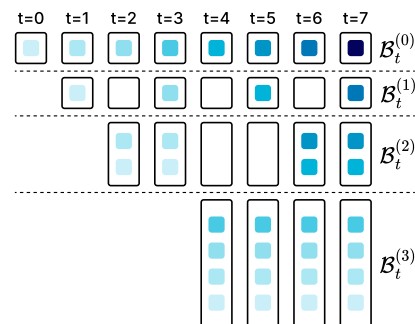

From a decoding perspective, attention can be viewed as a mechanism that partitions the prefix $[0, t)$ into a set of buckets, each summarizing a portion of the past. In vanilla attention, every token forms its own bucket, resulting in $t$ buckets of size 1, each stored as a fixed-size state (the KV caches. At the other extreme, linear attention (and state-space models) aggregates the entire prefix into a single bucket of size $t$, again represented by a fixed-size state.

Log-linear attention strikes a balance by partitioning the prefix into buckets of exponentially increasing size via a Fenwick-tree decomposition (Ryabko, 1992; Fenwick, 1994). This induces a natural inductive bias: recent tokens are retained at high resolution, while more distant tokens

**Figure 2:** Fenwick tree bucket assignments.

are summarized more coarsely. The partition contains at most $L = \mathcal{O}(\log T)$ disjoint buckets indexed by level $\ell$.[8] Each bucket $\mathcal{B}_t^{(\ell)}$ has size $|\mathcal{B}_t^{(\ell)}| = 2^{\ell-1}$ for $\ell \geq 1$, plus a sentinel bucket $\mathcal{B}_t^{(0)}$ of size 1. See Fig. 2 for an illustration.

---

[7]This algorithm depends on the chunk size $C$, but since $C$ is a hyperparameter this is still linear in $T$.

[8]More precisely, this divides the prefix $[0, t)$ into up to $L = \lceil \log_2 t + 1 \rceil + 1$ disjoint buckets. This decomposition is guided by the function $\text{lssb}(t) = \max\{\ell \in \mathbb{N} \mid 2^\ell \text{ divides } t\}$, which identifies the least significant set bit in the binary representation of $t$. Conceptually, the partitioning proceeds greedily, at each step

Log-linear attention maintains a separate recurrent memory $\mathbf{S}_t^{(\ell)} \in \mathbb{R}^{d \times d}$ for each bucket. At time $t$, the contribution of bucket $\ell$ to the output is weighted by a nonnegative coefficient $\lambda_t^{(\ell)}$, parameterized as a linear function of the current input $\boldsymbol{x}_t$. This allows the model to adaptively emphasize different temporal scales. The output is computed as,

$$\boldsymbol{o}_t = \sum_{\ell=0}^{L-1} \lambda_t^{(\ell)} \boldsymbol{q}_t^\top \left( \sum_{s \in \mathcal{B}_t^{(\ell)}} \boldsymbol{v}_s \boldsymbol{k}_s^\top \right) = \sum_{\ell=0}^{L-1} \lambda_t^{(\ell)} \boldsymbol{q}_t^\top \mathbf{S}_t^{(\ell)}. \tag{3}$$

We observe that when all $\lambda_t^{(\ell)}$ are the same (or more generally when the $\lambda_t^{(\ell)}$ and $\lambda_t^{(\ell')}$ are linearly related across time) log-linear attention collapses to linear attention. Allowing distinct $\lambda_t^{(\ell)}$ is therefore essential for capturing multi-scale temporal structure.

**Parallel form.** The recurrent form in Eq. 3 is conceptually simple but inefficient on modern accelerators, which are optimized for high-throughput matrix–matrix multiplication. To leverage this hardware and enable parallelization across time, we reformulate the expression in a matrix-multiplication–friendly form as in §2:

$$\mathbf{O} = \underbrace{\left( \mathbf{Q}\mathbf{K}^\top \odot \mathbf{M}^{\mathcal{H}} \right)}_{\mathbf{P}} \mathbf{V}, \quad \mathbf{M}_{ts}^{\mathcal{H}} = \begin{cases} \lambda_t^{\ell(t,s)} & \text{if } s \leq t, \\ 0 & \text{otherwise,} \end{cases} \tag{4}$$

where $\ell(t, s)$ denotes the bucket level of token $s$ relative to time $t$ under Fenwick-tree partitioning. For readability, we omit explicit $(t, s)$ indices when unambiguous. The matrix $\mathbf{P}$ is a hierarchical matrix which inherits structured low-rank pattern from the hierarchical partitioning, given below. In §3.3, we exploit this structure to design a parallel training algorithm with $\mathcal{O}(T \log T)$ complexity.

$$\begin{bmatrix} \begin{matrix} \lambda_0^{(0)} \boldsymbol{q}_0^\top \boldsymbol{k}_0 \\ \lambda_1^{(1)} \boldsymbol{q}_1^\top \boldsymbol{k}_0 \quad \lambda_1^{(0)} \boldsymbol{q}_1^\top \boldsymbol{k}_1 \end{matrix} & & & \\ \begin{matrix} \begin{bmatrix} \lambda_2^{(2)} \boldsymbol{q}_2 \\ \lambda_3^{(2)} \boldsymbol{q}_3 \end{bmatrix} \begin{bmatrix} \boldsymbol{k}_0 \\ \boldsymbol{k}_1 \end{bmatrix}^\top \end{matrix} & \begin{matrix} \lambda_2^{(0)} \boldsymbol{q}_2^\top \boldsymbol{k}_2 \\ \lambda_3^{(1)} \boldsymbol{q}_3^\top \boldsymbol{k}_2 \quad \lambda_3^{(0)} \boldsymbol{q}_3^\top \boldsymbol{k}_3 \end{matrix} & & \\ \begin{matrix} \begin{bmatrix} \lambda_4^{(3)} \boldsymbol{q}_4 \\ \lambda_5^{(3)} \boldsymbol{q}_5 \\ \lambda_6^{(3)} \boldsymbol{q}_6 \\ \lambda_7^{(3)} \boldsymbol{q}_7 \end{bmatrix} \begin{bmatrix} \boldsymbol{k}_0 \\ \boldsymbol{k}_2 \\ \boldsymbol{k}_3 \\ \boldsymbol{k}_1 \end{bmatrix}^\top \end{matrix} & \begin{matrix} \lambda_4^{(0)} \boldsymbol{q}_4^\top \boldsymbol{k}_4 \\ \lambda_5^{(1)} \boldsymbol{q}_5^\top \boldsymbol{k}_4 \quad \lambda_5^{(0)} \boldsymbol{q}_5^\top \boldsymbol{k}_5 \\ \begin{bmatrix} \lambda_6^{(2)} \boldsymbol{q}_6 \\ \lambda_7^{(2)} \boldsymbol{q}_7 \end{bmatrix} \begin{bmatrix} \boldsymbol{k}_4 \\ \boldsymbol{k}_5 \end{bmatrix}^\top \end{matrix} & \begin{matrix} \lambda_6^{(0)} \boldsymbol{q}_6^\top \boldsymbol{k}_6 \\ \lambda_7^{(1)} \boldsymbol{q}_7^\top \boldsymbol{k}_6 \quad \lambda_7^{(0)} \boldsymbol{q}_7^\top \boldsymbol{k}_7 \end{matrix} \end{bmatrix}$$

**Remark.** The matrix $\mathbf{M}^{\mathcal{H}}$ (and $\mathbf{A}$) is a lower-triangular instance of a hierarchical ($\mathcal{H}$) matrix—specifically, of the HODLR (Hierarchically Off-Diagonal Low-Rank) type. When constructed using schemes like the Fenwick tree, it inherits the recursive partitioning and low-rank off-diagonal blocks that define $\mathcal{H}$ matrices. This establishes a direct connection between log-linear attention and hierarchical matrices: the attention operator corresponds to structured matrix multiplication with an $\mathcal{H}$ matrix. We refer to $\mathbf{M}^{\mathcal{H}}$ as a quasi-$\mathcal{H}$ matrix—a specialized class lying between general $\mathcal{H}$ and semiseparable matrices, designed to support $\mathcal{O}(\log T)$-space recurrence. See Section B.1 for details.

## 3.2 MEMORY-EFFICIENT DECODING

Let $\text{lssb}(t)$ denote the index of the least significant set bit in the binary representation of $t$. The states $\{\mathbf{S}_t^{(\ell)}\}_\ell$ evolve according to the following recurrence (using linear attention for simplicity):

---

subtracting the largest power of two that fits within the remaining segment of the prefix,

$$b_t^{(i)} = \begin{cases} t & \text{if } i = 0 \\ b_t^{(i-1)} - 2^{\text{lssb}\left(b_t^{(i-1)}\right)} & \text{otherwise} \end{cases}, \quad \mathcal{B}_t^{(\ell)} = \begin{cases} \{b_t^{(0)}\} & \text{if } \ell = 0 \\ \{b_t^{(i+1)}, \cdots, b_t^{(i)} - 1\} & \text{if } \ell = \text{lssb}\left(b_t^{(i)}\right) + 1 \\ \varnothing & \text{otherwise} \end{cases}$$

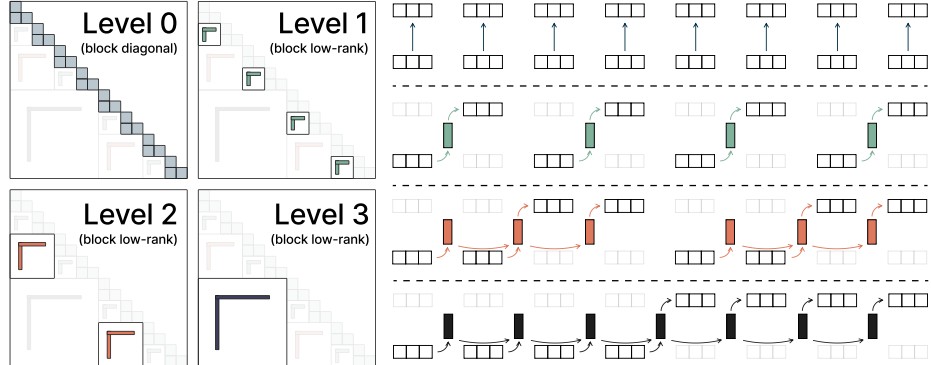

**Figure 3: Left**: Decomposition of the matrix $\mathbf{M}^{\mathcal{H}}$. **Right**: Chunkwise algorithm (Algorithm 1). Level 0 handles intra-chunk computations using a quadratic (in chunk size) algorithm, which is efficient due to small chunk sizes. Levels 1 and above perform inter-chunk computations by invoking existing inter-chunk primitives multiple times, with overall complexity logarithmic in the number of chunks.

$$\mathbf{S}_t^{(\ell)} = \begin{cases} \boldsymbol{v}_t \boldsymbol{k}_t^\top & \text{if } \ell = 0 \\ 0 & \text{if } 0 < \ell \le \mathrm{lssb}(t) \\ \sum_{\ell'=0}^{\ell-1} \mathbf{S}_{t-1}^{(\ell')} & \text{if } \ell = \mathrm{lssb}(t)+1 \\ \mathbf{S}_{t-1}^{(\ell)} & \text{if } \ell > \mathrm{lssb}(t)+1 \end{cases}$$

At each step, the immediate term $\boldsymbol{v}_t \boldsymbol{k}_t^\top$ enters the finest level; buckets up to $\mathrm{lssb}(t)$ merge and promote one level coarser. When $t$ is a power of two the hierarchy expands by one bucket. This Fenwick-like organization enables online processing with $\mathcal{O}(\log T)$ memory while retaining efficient multiscale access.

### 3.3 EFFICIENT ALGORITHM FOR TRAINING

Chunkwise parallelism for linear attention (Sun et al., 2023; Yang et al., 2023; Dao & Gu, 2024) partitions a sequence of length $T$ into chunks of size $C$, which are processed in parallel while exchanging only limited information across boundaries. This approach balances two extremes: it avoids the prohibitive cost of global attention while exposing substantially more parallelism than purely recurrent execution. We extend this idea to the log-linear setting and develop an efficient *chunkwise* training algorithm.

For a given chunk size $C$, the matrix $\mathbf{M}^{\mathcal{H}}$ admits the structured decomposition,

$$\mathbf{M}^{\mathcal{H}} = \mathbf{D} + \sum_{\ell=\ell_C}^{L-1} \mathbf{M}^{(\ell)}, \quad \mathbf{M}_{ts}^{(\ell)} = \begin{cases} \lambda_t^{(\ell)} \mathbf{M}_{ts}^{\mathcal{S}}, & \text{if } s \in \mathcal{B}_t^{(\ell)}, \\ 0, & \text{otherwise.} \end{cases} \tag{5}$$

where $\mathbf{D}$ is block-diagonal with $\frac{T}{C}$ causal blocks $\{\mathbf{D}^{[k]}\}$ of size $(C \times C)$, capturing intra-chunk interactions via $(\mathbf{D}^{[i]})_{ts} = \lambda_{iC+t}^{(\ell)} \mathbf{M}_{ts}^{\mathcal{S}}$. The remaining $\{\mathbf{M}^{(\ell)}\}$ encode inter-chunk dependencies in blockwise low-rank form. Indexing begins at $\ell_C$, the level aligned to chunk size $C$; levels $\ell < \ell_C$ collapse into $\mathbf{D}$ (Fig. 3, left).

Building on this structure, we propose a chunkwise algorithm for log-linear attention (Algorithm 1). As summarized in Fig. 3 (right), the method introduces only a logarithmic overhead compared with standard linear attention. Computation proceeds in two stages:

**Intra-chunk stage** ($\ell < \ell_C$). The block-diagonal component $\mathbf{D}$ is treated as a dense matrix within each chunk. Each block costs $\mathcal{O}(C^2)$, giving a total complexity of $\mathcal{O}(TC)$.

**Inter-chunk stage** ($\ell \ge \ell_C$). The matrices $\{\mathbf{M}^{(\ell)}\}$ reduce to scaled sequentially semi-separable structures (Eq. 5). With efficient state-passing primitives (e.g., Mamba-2, Gated DeltaNet), inter-chunk dependencies are computed using only $\mathcal{O}\left(\log \frac{T}{C}\right)$ primitive calls. Each call requires $\mathcal{O}(T)$ time and memory,[9] leading to an overall complexity of $\mathcal{O}(T \log \frac{T}{C})$.

Our algorithm extends the classical parallel prefix-sum (scan) to a hierarchical setting—a *chunkwise parallel scan*. Unlike token-level scans, which often suffer from memory-bandwidth bottlenecks

---

[9]At level $\ell$, $\mathbf{M}^{(\ell)}$ contains $\frac{T}{2^{\ell-1}C}$ chunks of size $2^{\ell-1}C$. Redundant work can be avoided, reducing cost by a constant factor of two.

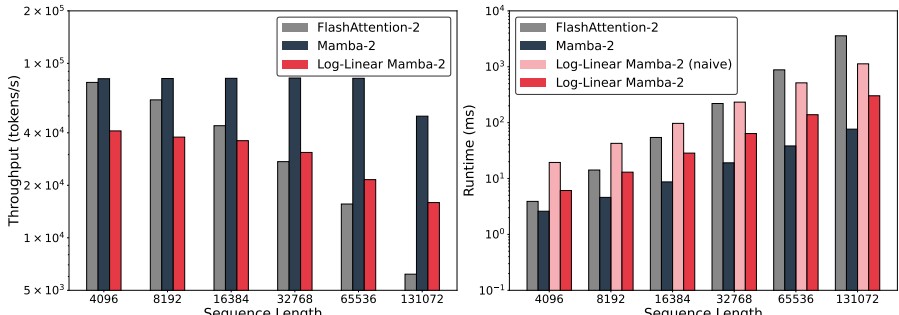

**Figure 4:** Training throughput (left; higher is better) and kernel runtime for a forward and backward pass (right; lower is better) across varying sequence lengths. **Log-Linear Mamba-2 (naive)** denotes repeated application of the existing Mamba-2 primitives, while **Log-Linear Mamba-2** uses a custom implementation with optimizations such as level fusion. The throughput drop at sequence length 131K is due to gradient checkpointing to reduce memory usage. Experiments were run on an H100 GPU with batch size 2, 48 heads, head dimension 64, state dimension 128, and chunk size 64. We use MVA for (Log-Linear) Mamba-2, and GQA for FlashAttention-2.

during training (Yang et al., 2023), the chunkwise formulation reorganizes recurrent updates into parallel chunk operations. Concretely, it executes $\mathcal{O}(\log T)$ independent scans (one per memory level), each implementable with standard methods such as the Blelloch scan (Blelloch, 1990). Layer-specific weights (e.g., $\lambda_t^{(\ell)}$) can easily be incorporated into these scans.

### 3.4 Log-Linear Variants of Mamba-2 and Gated DeltaNet

We next apply the above construction to Mamba-2 Dao & Gu (2024) and Gated DeltaNet Yang et al. (2024a). As discussed in §2, both models use gating mechanisms that induce a sequentially semiseparable (SSS) temporal structure in the mask $\mathbf{M}^{\mathcal{S}}$ (with $\mathbf{M}_{ij} = \prod_{k=j+1}^{i} \alpha_k$; see Eq. 2). The two architectures differ in how they parameterize the transition matrix $\mathbf{A}$.

Our approach preserves the original form of $\mathbf{A}$ in each model while composing the attention mask with its log-linear variant $\mathbf{M} = \mathbf{M}^{\mathcal{S}} \odot \mathbf{M}^{\mathcal{H}}$.[10] We refer to the resulting models as *log-linear* Mamba-2 and *log-linear* Gated DeltaNet. Their parallel forms are given by,

$$\mathbf{O} = \left(\mathbf{Q}\mathbf{K}^T \odot \mathbf{M}^{\mathcal{S}} \odot \mathbf{M}^{\mathcal{H}}\right) \mathbf{V} \qquad \qquad \textit{Log-Linear Mamba-2}$$

$$\mathbf{O} = \left(\left(\mathbf{Q}\mathbf{K}^{\top} \odot \mathbf{L}\right)\left(\mathbf{I} + \mathbf{K}\mathbf{K}^{\top} \odot (\mathbf{L} - \mathbf{I})\right)^{-1} \odot \mathbf{M}^{\mathcal{S}} \odot \mathbf{M}^{\mathcal{H}}\right) \mathbf{V} \quad \textit{Log-Linear Gated DeltaNet}$$

More broadly, any linear-attention mechanism with structured memory and an efficient chunkwise-parallel primitive can be "lifted" to a log-linear variant by composing its temporal mask with $\mathbf{M}^{\mathcal{H}}$.

### 3.5 Implementation

We implemented the chunkwise parallel scan algorithm in `Triton` (Tillet et al., 2019). The custom kernel for log-linear Mamba-2 outperforms FlashAttention-2 (Dao, 2024) (forward + backward) at sequence lengths beyond 8K. In full training setups, throughput depends on model architecture. Notably, log-linear Mamba-2 (with MLP) surpasses Transformer throughput at 32K, despite additional layers like depthwise convolutions absent in the Transformer. See Fig. 4 and Sec. C for details.

## 4 Experiments

We conduct a suite of experiments across both synthetic and real-world benchmarks. We emphasize that our experiments are not necessarily intended position log-linear attention as the best subquadratic architecture, but rather to highlight the promise of our framework compared to sensible baselines.

### 4.1 Synthetic Benchmark

---

[10]More precisely, the elementwise product of an SSS matrix and an $\mathcal{H}$ matrix remains an $\mathcal{H}$ matrix. We separate them here for clarity.

We begin by evaluating models on the multi-query associative recall (MQAR) task (Arora et al., 2023), a standard diagnostic benchmark for testing in-context recall. Our setup closely follows Arora et al. (2024): models are trained and evaluated on 256-token sequences containing 4 to 64 key-value pairs (excluding the length generalization component), with tuned learning rates. For log-linear models, we also tune the $\lambda$ parameterization.

| Dimension | 16 | 32 | 64 |
|---|---|---|---|
| Transformer | $\geq 99$ | $\geq 99$ | $\geq 99$ |
| Mamba-2 | 46.9 (2.3) | 75.1 (4.9) | 89.6 (6.1) |
| w/ *Log-Linear* | 55.9 (9.1) | 76.5 (4.8) | 92.9 (2.7) |
| Gated DeltaNet | 38.4 (1.0) | 79.0 (2.1) | $\geq 99$ |
| w/ *Log-Linear* | 40.0 (1.4) | 84.4 (1.2) | $\geq 99$ |

**Table 2:** Average accuracies and standard deviations (in parentheses) on MQAR over 5 seeds. Training was early stopped when accuracy exceeded 99%.

We run each configuration with five seeds. Training was early stopped when accuracy exceeded 99%. Additional experimental details are provided in §D. As shown in Table 2, log-linear attention performs well—even when applied on top of associative recall-optimized models like Gated DeltaNet.

## 4.2 LANGUAGE MODELING

We perform academic-scale language modeling pretraining from scratch using 50B tokens on the Long-Data-Collections dataset,[11] using a sequence length of 16K. All models have 21 layers and use a hidden size of 1536. We use a Transformer with 16 attention heads and a RoPE base of 500K, a modified Mamba-2 with 48 heads and MLP layers, and a Gated DeltaNet with 6 heads. The Transformer, Mamba-2, and Gated DeltaNet models contain 693M, 802M, and 793M parameters, respectively. For the *log-linear* variants, we apply a linear layer on top of the hidden states to compute the per-head values $\lambda_t^{(\ell)}$. This adds less than 3% additional parameters for Mamba-2 (825M) and less than 0.4% for Gated DeltaNet (796M). Since Mamba-2 and Gated DeltaNet have more parameters than ordinary Transformers, we also include a (roughly) parameter-matched Transformer variant with 24 layers (778M parameters) for comparison. For our log-linear variants, we use the default hyperparameters from the baselines (§D). We also evaluated a parameter-matched Hyena model Poli et al. (2023), which also has log-linear compute (but linear memory). As its WikiText perplexity (around 29) was substantially higher than that of the other models ($<23$), our main experiments focus on the Transformer, Mamba-2, and Gated DeltaNet families.

**Standard benchmarks.** Following prior work (Dao & Gu, 2024; Yang et al., 2024a), we evaluate models on WikiText perplexity and several zero-shot commonsense reasoning benchmarks (Table 6). These are short-context tasks and are therefore largely insensitive to model state size. As such, we generally expect the log-linear variants to perform comparably to their linear counterparts. Log-Linear Mamba-2 improves upon its linear counterpart in perplexity and in half of the commonsense

| Model | Wiki. ppl ↓ | LMB. ppl ↓ | LMEval average ↑ |
|---|---|---|---|
| Transformer | 21.56 | 22.14 | 44.0 |
| w/ *24 Layers* | 21.13 | 21.17 | 45.6 |
| Hyena | 29.50 | / | / |
| Mamba-2 | 22.44 | 24.14 | 44.8 |
| w/ *Log-Linear* | 22.11 | 21.86 | 44.9 |
| Gated DeltaNet | 21.73 | 19.71 | 45.0 |
| w/ *Log-Linear* | 21.45 | 18.09 | 45.5 |

**Table 3:** PPL and commonsense reasoning.

reasoning tasks. Log-Linear Gated DeltaNet shows stronger gains, outperforming its linear version in perplexity and in all but one reasoning benchmark. Notably, it also outperforms a layer-matched Transformer across all metrics and a parameter-matched Transformer on half of them.

**Per-position loss.** Following Lin et al. (2025), we report the model's loss at each token position to evaluate its ability to handle long contexts (Fig. 5). If the loss steadily decreases as the token position increases, it suggests the model is effectively using the full context. However, if the loss levels off after a certain point, it indicates the model struggles to make use of information that is too far back in the sequence. For this analysis, we use 39M tokens from Book-3.[12] To improve visualization, we apply a running average with a window size of 501. We observe that extending both Mamba-2 and Gated DeltaNet to their log-linear counterparts consistently reduces the (smoothed) loss across various positions, indicating improved long-range context utilization. Log-Linear Gated DeltaNet

---

[11]https://huggingface.co/datasets/togethercomputer/Long-Data-Collections.
[12]victor-wu/book3

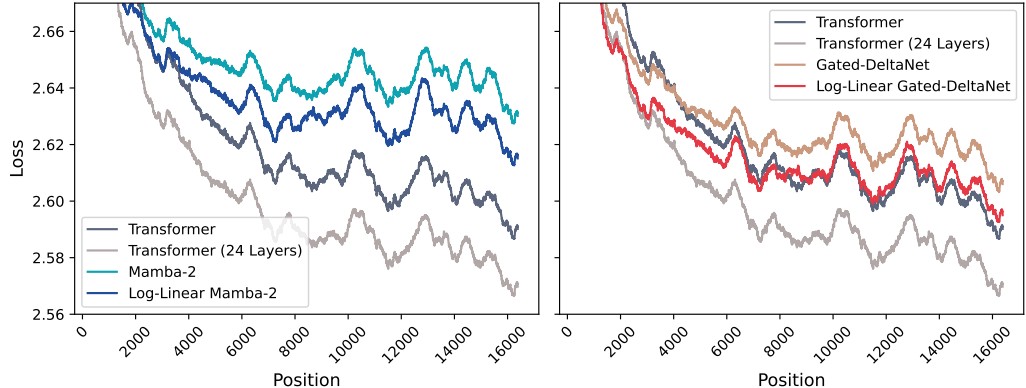

**Figure 5:** Per-position loss on Book3 samples (about 39M tokens) with running average of window size 501.

| Model | S-NIAH-1 (pass-key retrieval) | | | S-NIAH-2 (number in haystack) | | | S-NIAH-3 (uuid in haystack) | | |
|---|---|---|---|---|---|---|---|---|---|
| | 4K | 8K | 16K | 4K | 8K | 16K | 4K | 8K | 16K |
| Transformer | 72.6 | 76.0 | 16.6 | 100.0 | 99.8 | 90.0 | 77.4 | 67.0 | 44.6 |
| w/ *24 Layers* | 92.4 | 78.4 | 89.8 | 100.0 | 100.0 | 99.6 | 84.0 | 63.6 | 36.4 |
| Mamba-2 | 90.4 | 56.8 | 21.6 | 72.4 | 28.0 | 18.6 | 4.0 | 3.6 | 0.8 |
| w/ *Log-Linear* | 100.0 | 99.8 | 72.4 | 89.8 | 68.2 | 12.8 | 33.6 | 22.6 | 2.0 |
| Gated DeltaNet | 100.0 | 100.0 | 100.0 | 95.8 | 46.8 | 5.0 | 66.2 | 14.6 | 6.0 |
| w/ *Log-Linear* | 100.0 | 100.0 | 100.0 | 95.6 | 59.6 | 9.2 | 48.8 | 13.0 | 8.8 |

| Model | MK-NIAH-1 (multi-key line retrieval) | | | MQ-NIAH (multi-query) | | | MV-NIAH (multi-value) | | |
|---|---|---|---|---|---|---|---|---|---|
| | 4K | 8K | 16K | 4K | 8K | 16K | 4K | 8K | 16K |
| Transformer | 79.4 | 83.0 | 61.4 | 58.9 | 48.0 | 29.8 | 37.5 | 34.1 | 21.5 |
| w/ *24 Layers* | 62.6 | 83.2 | 75.2 | 54.6 | 46.0 | 34.5 | 48.4 | 45.4 | 32.3 |
| Mamba-2 | 27.2 | 18.6 | 13.6 | 28.7 | 19.4 | 1.3 | 27.9 | 14.8 | 4.4 |
| w/ *Log-Linear* | 43.2 | 39.8 | 21.2 | 26.6 | 22.4 | 6.6 | 28.1 | 22.8 | 8.9 |
| Gated DeltaNet | 23.0 | 21.2 | 5.2 | 21.6 | 16.9 | 7.2 | 16.2 | 14.5 | 7.0 |
| w/ *Log-Linear* | 49.4 | 27.8 | 10.2 | 34.9 | 22.0 | 9.8 | 31.4 | 25.0 | 13.3 |

**Table 4:** NIAH experiments with three single/multi-needle tasks.

also closely tracks the performance of the layer-matched Transformer, although a performance gap remains when compared to the parameter-matched Transformer.

**Needle-In-A-Haystack.** We use the Needle-In-A-Haystack (NIAH, Table 4 and Fig. 10) benchmark from RULER (Hsieh et al., 2024), where the model must retrieve a value (the "needle") based on a key hidden in a long context (the "haystack"). In the simpler single-needle tasks, the log-linear variant of Mamba-2 outperformed its linear counterpart on 8 out of 9 metrics. Gated DeltaNet, which already achieved perfect accuracy in several cases, saw improvements in 3 metrics, with 3 remaining unchanged. For the more challenging multi-needle tasks, Log-Linear Mamba-2 again improved in 8 out of 9 metrics, while Log-Linear Gated DeltaNet achieved improvements across all metrics.

**Other tasks.** Due to space we show the results on the in-context retrieval benchmark (Arora et al., 2023) and LongBench (Bai et al., 2023) in the appendix.

## 5 DISCUSSION AND LIMITATIONS

While log-linear attention improves upon linear attention in many cases, there are still quite a few tasks where it did not improve upon the linear attention baselines. Due to compute resources we were unable to experiment with different parameterizations of the $\lambda$ terms (or hyperparameters in general),[13] and it is possible that optimal parameterization of $\lambda$ could lead to improved results. We also still observe a significant performance gap compared to Transformers across all benchmarks.

---

[13]We were only able to run our 700M-800M parameter language models just once due to compute constraints.

The engineering complexity of log-linear attention is higher. Inter-chunk computations conceptually resemble multiple applications of linear attention primitives, but intra-chunk operations require bespoke implementations. These intra-chunk mechanisms are a primary factor behind the speed differences. Additionally, the backward pass is more intricate, as it requires (manually) computing the gradients not only for the standard attention components but also for the additional $\lambda$ terms.

The use of Fenwick-tree partitioning (§3.1) introduces an inductive bias: recent tokens are allocated more fine-grained memory, while distant tokens are compressed more aggressively. This design reflects a natural assumption rooted in hierarchical matrix which posits that interactions between distant elements can be approximated in low-rank form. While intuitive and inspired by physical phenomena, this inductive bias may not be optimal for all applications. Future work could explore extensions that enable more flexible structures while preserving computational efficiency.

Finally, in this work, we extended two existing linear-attention/SSM architectures to their log-linear counterparts, namely Mamba-2 (Dao & Gu, 2024) and Gated DeltaNet (Yang et al., 2024a). Several other promising architectures, including xLSTM (Beck et al., 2024; 2025b) and MesaNet (Von Oswald et al., 2023; von Oswald et al., 2025), may likewise benefit from log-linear formulations. Developing and evaluating log-linear variants of these models is an exciting direction for future research.

## 6 RELATED WORK

**Structured matrices for deep learning architectures.** Modern architectures combine token- and channel-mixing layers, both based on matrix multiplications. Recent work replaces dense layers with *structured matrices*. For channel mixing, approaches include Butterfly (Dao et al., 2020), Monarch matrices (Dao et al., 2022a), and more recently, Block Tensor-Train matrices (Qiu et al., 2024). Token mixing has been exemplified by the family of linear attention models (Katharopoulos et al., 2020) and their various kernelizations (Xiong et al., 2021). Dao & Gu (2024) generalize these approaches by extending low-rank structures to semiseparable matrices, enabling efficient recurrent inference and subsuming many recent recurrent models. Another line uses sparse patterns like sliding-window attention, alongside several hybrid methods (Nguyen et al., 2021; Arora et al., 2025; Munkhdalai et al., 2024).

**Log-linear complexity sequence modeling.** Several prior efforts have focused on reducing the quadratic cost of attention to log-linear time complexity (Kitaev et al., 2020; Shi et al., 2023; Cunningham et al., 2024; Qin et al., 2023; Fu et al., 2023; Madaan et al.; Ye et al., 2019). Approaches such as LogSparse Transformer (Li et al., 2019) and Informer (Zhou et al., 2021) introduce sparse attention patterns to improve computational efficiency, particularly in time-series applications. Reformer (Kitaev et al., 2020) employs locality-sensitive hashing (LSH) to efficiently cluster similar queries and keys. Multi-resolution attention (Zeng et al., 2022) adopts a hierarchical approach, progressively refining attention scores from coarse to fine granularity, while Fast Multipole Attention (Kang et al., 2024) adapts the classical fast multipole method to efficiently model long-range interactions. A similar viewpoint connects log-linear attention to dilated convolution (Van Den Oord et al., 2016) through their hierarchical mixing structure. Dilated convolution extends convolution, which corresponds to Toeplitz matrices, whereas we operate primarily with semi-separable and hierarchical matrices. In our work, we leverage the Fenwick tree data structure—a specialized binary indexed tree that enables efficient prefix sum calculations and updates in logarithmic time—to design an efficient attention layer during both training and decoding phases. While Zhu & Soricut (2021) also employ hierarchical matrices for attention, their formulation is fully parallel and targeted at modest sequence lengths. In contrast, our approach adopts a chunkwise-parallel strategy with a custom Triton implementation optimized for long-sequence training. Concurrently, Yau et al. (2025) propose a related architecture with $\mathcal{O}(\log T)$ memory, using a relaxed prefix-scan algorithm for state aggregation that accommodates arbitrary (potentially non-associative) functions.

## 7 CONCLUSION

We introduced Log-Linear Attention, a general framework that extends a broad class of linear attention and state-space models to their log-linear counterparts—models with logarithmically growing state size. This framework offers both theoretical insights and practical benefits, linking structured matrix theory with hardware-efficient computation. As a case study, we applied this approach to two recent architectures: Mamba-2 and Gated DeltaNet.

ACKNOWLEDGMENTS

We thank Tianyuan Zhang, Jyothish Pari, Adam Zweiger, and Yu Zhang for helpful discussion. This study was supported by the MIT-Google Program for Computing Innovation, MIT-IBM Watson AI Lab, and the AI2050 program at Schmidt Sciences (Grant G-25-67980). HG was supported by a Microsoft PhD Fellowship.

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

| Model | Temporal Structure | Hidden Size Structure |
|---:|---|---|
| Mamba-2 | Semiseparable | Scaled Identity |
| Gated DeltaNet | Semiseparable | Identity plus Low-Rank |
| *Log-Linear* Mamba-2 | Hierarchical | Scaled Identity |
| *Log-Linear* Gated DeltaNet | Hierarchical | Identity plus Low-Rank |

**Table 5:** Structural comparison of different attention variants.

## A  GENERALIZING LOG-LINEAR ATTENTION TO MORE EXPRESSIVE LINEAR RNNS

The main paper adopts the following unified view of efficient attention (Eq. 1):

$$\mathbf{P} = \mathbf{A} \odot \mathbf{M}, \quad \mathbf{O} = \mathbf{P}\mathbf{V},$$

This formulation reveals that the key difference between linear and log-linear attention lies in the structure of the mask matrix $\mathbf{M} \in \mathbb{R}^{T \times T}$. Variations among linear attention models—such as Mamba-2 and Gated DeltaNet—stem from different parameterizations of $\mathbf{A}$. While this perspective offers a unifying and intuitive framework that captures a wide range of attention mechanisms, it comes with an important limitation: the state-transition terms are restricted to be scalars (in the case of Mamba-2) or identity-plus-rank-one matrices (in the case of Gated DeltaNet).

In this section, we introduce a more general framework that relaxes this scalar constraint by allowing state-transition terms (including the thus $\lambda_t^{(\ell)}$ terms) to be matrix-valued. This extension enables richer and more expressive attention mechanisms while preserving computational efficiency.

**Linear Attention as an SSS Tensor.** Consider the standard linear attention mechanism with data-dependent gating and an SSS (sequentially semiseparable) mask $\mathbf{M}^{\mathcal{S}}$:

$$\mathbf{P} = \mathbf{Q}\mathbf{K}^\top \odot \mathbf{M}^{\mathcal{S}}, \quad \mathbf{O} = \mathbf{P}\mathbf{V}.$$

In the main paper, we extend the SSS mask $\mathbf{M}^{\mathcal{S}}$ to a hierarchical form $\mathbf{M}^{\mathcal{H}}$. Notice that in Mamba-2, the resulting matrix $\mathbf{P}$ also inherits the same structural property, with its SSS-rank governed by the hidden dimension $d$:

$$\mathbf{P}_{t,s} = \mathbf{Q}_t \left( \mathbf{C}_t \cdots \mathbf{C}_{s+1} \right) \mathbf{K}_s^\top, \quad \text{where} \quad \mathbf{C}_t = \alpha_t \mathbf{I}.$$

We now define a 4D tensor $\mathbf{\mathsf{M}}^{\mathcal{S}} \in \mathbb{R}^{(T \times T) \times (d \times d)}$ such that:

$$\mathbf{P}_{t,s} = \mathbf{Q}_t \mathbf{\mathsf{M}}_{t,s} \mathbf{K}_s^\top, \quad \text{where} \quad \mathbf{\mathsf{M}}_{t,s} = \mathbf{C}_t \cdots \mathbf{C}_{s+1}.$$

Each entry $\mathbf{\mathsf{M}}_{t,s} \in \mathbb{R}^{d \times d}$ is a matrix, making $\mathbf{\mathsf{M}}^{\mathcal{S}}$ a 4D tensor. We refer to this as an SSS tensor due to its sequentially semiseparable-like structure along the temporal dimension, though this term is not yet formalized in the literature.

This tensor-centric view naturally accommodates matrix-valued state transitions $\mathbf{C}_t \in \mathbb{R}^{d \times d}$ with arbitrary structure, offering a richer representation than scalar- or identity-plus-rank-one-based approaches. In particular, models such as Mamba-2 and Gated DeltaNet can be interpreted as operating on 4D tensors with different hidden-dimension structures, while still preserving temporal semiseparability.[14]

$$\text{Mamba-2:} \quad \mathbf{\mathsf{M}}_{t,s}^{\mathcal{S}} = \prod_{t'=t}^{s+1} \alpha_{t'} \mathbf{I}, \qquad \text{Gated DeltaNet:} \quad \mathbf{\mathsf{M}}_{t,s}^{\mathcal{S}} = \prod_{t'=t}^{s+1} \alpha_{t'} \left( \mathbf{I} - \beta_{t'} \boldsymbol{k}_{t'} \boldsymbol{k}_{t'}^\top \right)$$

**Log-Linear Attention as an $\mathcal{H}$ Tensor.** We can apply our *log-linear* attention to these more flexible (linear) RNNs by incorporating matrix-valued, level- and data-dependent terms $\mathbf{\Lambda}_t^{(\ell)} \in \mathbb{R}^{d \times d}$:

$$\text{Mamba-2:} \quad \mathbf{\mathsf{M}}_{t,s}^{\mathcal{H}} = \mathbf{\Lambda}_t^{(\ell)} \prod_{t'=t}^{s+1} \alpha_{t'} \mathbf{I}, \qquad \text{Gated DeltaNet:} \quad \mathbf{\mathsf{M}}_{t,s}^{\mathcal{H}} = \mathbf{\Lambda}_t^{(\ell)} \prod_{t'=t}^{s+1} \alpha_{t'} \left( \mathbf{I} - \beta_{t'} \boldsymbol{k}_{t'} \boldsymbol{k}_{t'}^\top \right)$$

---

[14]Strictly speaking, Gated DeltaNet also need to include a term $\beta_t$ from $\beta_t \boldsymbol{v}_t \boldsymbol{k}_t^\top$. For clarity, we omit it here, as it can be absorbed into other terms.

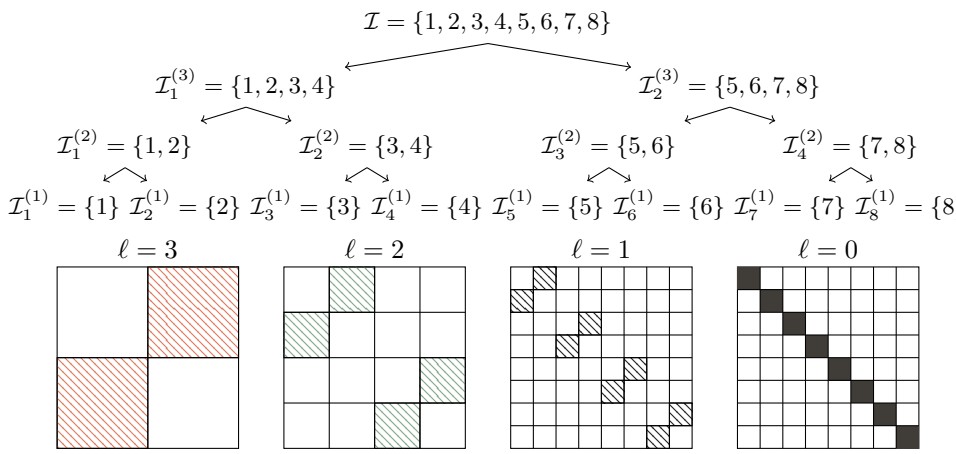

**Figure 6:** Visualization adapted from Massei et al. (2020); Kressner et al. (2019): This example illustrates a cluster tree of depth 3 along with the corresponding block partitions at each level. Blocks marked with stripes are stored as low-rank matrices in the HODLR format, while those filled with solid color represent dense matrices.

This formulation highlights a key insight: both Mamba-2 and Gated DeltaNet share a common semiseparable structure in the temporal dimension, but differ in how they structure the hidden dimension. Mamba-2 relies on scaled identities, while Gated DeltaNet applies identity-minus-rank-one modifications. Table 5 summarizes these distinctions.

# B  Log-Linear Attention as $\mathcal{H}$ Matrices

We begin by introducing two classes of Hierarchical matrices ($\mathcal{H}$ matrices) following Massei et al. (2020): HODLR (Hierarchically Off-Diagonal Low-Rank) matrices and HSS (Hierarchically Semi-Separable) matrices. We then show how Log-Linear Attention corresponds to a specific subclass of $\mathcal{H}$ matrices that occupies an intermediate position between these two. Finally, we discuss a further variant of $\mathcal{H}$ matrices that, in principle, allows for more refined partitioning—potentially enhancing approximation quality at the cost of increased (though constant-factor) computational complexity.

## B.1  HODLR Matrices

HODLR (Hierarchically Off-Diagonal Low-Rank) matrices are structured matrices built via recursive partitioning, where off-diagonal blocks are low-rank at every level. This structure is formalized using a cluster tree Massei et al. (2020). Let $T$ be the matrix dimension, and let $\mathcal{T}$ be a perfectly balanced binary tree of depth $L$ whose nodes are subsets of $\{1, \ldots, T\}$. We say $\mathcal{T}$ is a cluster tree if: (1) the root is $\mathcal{I} = \{1, \ldots, T\}$; (2) each level partitions indices into contiguous blocks; (3) every node $\mathcal{I}^{(\ell)} i$ at level $\ell$ has two children $\mathcal{I}_{2i-1}^{(\ell-1)}$ and $\mathcal{I}_{2i}^{(\ell-1)}$ that form a disjoint partition of the parent. See Fig. 6 for a visual example of such a hierarchical partitioning.

Now, let $\mathbf{M} \in \mathbb{R}^{T \times T}$ be a square matrix and $\mathcal{T}$ a cluster tree as described above. We say that $\mathbf{M}$ is a $(\mathcal{T}, k)$-HODLR matrix if,

$$\text{rank}\left(\mathbf{M}[\mathcal{I}_i^{(\ell)}, \mathcal{I}_j^{(\ell)}]\right) \leq k, \quad \forall\ \mathcal{I}_i^{(\ell)}, \mathcal{I}_j^{(\ell)} \in \text{siblings}\,(\mathcal{T})$$

This hierarchical low-rank structure enables efficient $\mathcal{O}(T \log T)$ storage and matrix-vector multiplication, making HODLR matrices a core component in fast algorithms for dense matrix computations. HODLR belongs to the broader class of rank-structured matrices known as Hierarchical matrices ($\mathcal{H}$ matrices).

## B.2  HSS Matrices

The $\mathcal{O}(T \log T)$ memory complexity of HODLR matrices arises from their recursive structure: they consist of $\mathcal{O}(\log T)$ levels, each storing low-rank factorizations that require $\mathcal{O}(T)$ space. In cases where these low-rank factors exhibit linear dependencies across levels, it is possible to exploit these relationships through nested hierarchical low-rank representations, potentially reducing the memory complexity to $\mathcal{O}(T)$ by eliminating the logarithmic factor Massei et al. (2020).

Let $\mathcal{I}_i^{(\ell)}$ and $\mathcal{I}_j^{(\ell)}$ denote a pair of sibling clusters at level $\ell$ in the cluster tree $\mathcal{T}$. Define $n^{(\ell)} = 2^{\ell-1}$ as the block size at level $\ell$. The off-diagonal block corresponding to these clusters can be parameterized as:

$$\mathbf{M}[\mathcal{I}_i^{(\ell)}, \mathcal{I}_j^{(\ell)}] = \mathbf{U}_i^{(\ell)} \mathbf{\Sigma}_{i,j}^{(\ell)} \left(\mathbf{V}_j^{(\ell)}\right)^\top, \quad \text{where} \quad \mathbf{U}_i^{(\ell)}, \mathbf{V}_j^{(\ell)} \in \mathbb{R}^{n^{(\ell)} \times k}, \ \mathbf{\Sigma}_{i,j}^{(\ell)} \in \mathbb{R}^{k \times k}$$

We call $\mathbf{M}$ matrix a Hierarchically Semiseparable matrices (HSS) if low-rank factors at different levels are linearly related through some "translation operators" $\mathbf{T}_{\mathbf{U}}^{(\ell)}, \mathbf{T}_{\mathbf{V}}^{(\ell)} \in \mathbb{R}^{2k \times k}$ such that,

$$\mathbf{U}_i^{(\ell)} = \begin{bmatrix} \mathbf{U}_{i_1}^{(\ell-1)} & 0 \\ 0 & \mathbf{U}_{i_2}^{(\ell-1)} \end{bmatrix} \mathbf{T}_{\mathbf{U},i}^{(\ell)}, \quad \mathbf{V}_j^{(\ell)} = \begin{bmatrix} \mathbf{V}_{j_1}^{(\ell-1)} & 0 \\ 0 & \mathbf{V}_{j_2}^{(\ell-1)} \end{bmatrix} \mathbf{T}_{\mathbf{V},j}^{(\ell)}$$

More broadly, HSS matrices belong to a subclass of $\mathcal{H}$ matrices known as $\mathcal{H}^2$ matrices.

## B.3 QUASI-HIERARCHICAL MATRIX.

As discussed above, when the low-rank basis matrices $\mathbf{U}^{(\ell)}$ and $\mathbf{V}^{(\ell)}$ exhibit linear relationships across levels $\ell$, the matrix $\mathbf{M}$ reduces to a semiseparable form. In this case, both storage and matrix-vector multiplication complexities can be reduced to $\mathcal{O}(T)$. Otherwise, $\mathbf{M}$ retains the general hierarchical structure with $\mathcal{O}(T \log T)$ complexity.

We define a *Quasi-Hierarchical Matrix* as one in which only one of the basis sequences, either $\mathbf{U}^{(\ell)}$ or $\mathbf{V}^{(\ell)}$, satisfies such a linear nesting property across levels, while the other does not. The matrix $\mathbf{M}^{\mathcal{H}}$ used in the Log-Linear model (Eq. 4) is an instance of this structure.

Both Hierarchical and Quasi-Hierarchical matrices incur $\mathcal{O}(T \log T)$ complexity for storage and computation during training. However, the use of Quasi-Hierarchical matrices plays a crucial role in enabling $\mathcal{O}(\log T)$ complexity during inference. We are not aware of a recurrent algorithm for general Hierarchical matrices that achieves logarithmic inference complexity.[15]

**Reparameterization.** More precisely, Eq. 4 represents a Quasi-Hierarchical matrix that has been specifically re-parameterized as a composition of the scalar weights $\lambda^{(\ell)}$ and a sequentially semiseparable (SSS) matrix $\mathbf{M}^{\mathcal{S}}$. This reparameterization serves two purposes: first, to highlight the connection between our use of $\mathcal{H}$ matrices and the SSS format adopted in prior work; and second, to enable the block decomposition into a hierarchy of SSS matrices, as shown in Eq. 5.

We present this re-parameterization below, along with its 4D tensor variant discussed in §A, where we additionally assume that the matrices $\mathbf{U}_i$ and $\mathbf{V}_j$ are invertible.

**Matrix:**

$$\mathbf{M}_{i,j}^{\mathcal{H}} := \tau_i^{(\ell)} u_i v_j \Leftrightarrow \lambda_i^{(\ell)} \prod_{t=j+1}^{i} \alpha_t$$

$$\Rightarrow \ \tau_i^{(\ell)} := \lambda_i^{(\ell)}, \ u_i := \prod_{t=0}^{i} \alpha_t, \ v_j := \prod_{t=0}^{j} \frac{1}{\alpha_t}$$

$$\Leftarrow \ \lambda_i^{(\ell)} := \tau_i^{(\ell)} u_i v_i, \quad a_t := \frac{r_{t-1}}{r_t}$$

**Tensor:**

$$\mathbf{M}_{i,j}^{\mathcal{H}} := \mathbf{T}_i^{(\ell)} \mathbf{U}_i \mathbf{V}_j^\top \Leftrightarrow \mathbf{\Lambda}_i^{(\ell)} \prod_{t=i}^{j+1} \mathbf{C}_t$$

$$\mathbf{T}_i^{(\ell)} := \mathbf{\Lambda}_i^{(\ell)}, \ \mathbf{U}_i := \prod_{t=i}^{0} \mathbf{C}_t, \ \mathbf{V}_j^\top := \prod_{t=0}^{j} \mathbf{C}_t^{-1}$$

$$\mathbf{\Lambda}_i^{(\ell)} := \mathbf{T}_i^{(\ell)} \mathbf{U}_i \mathbf{V}_i^\top, \quad \mathbf{C}_t := \mathbf{R}_t^{-1} \mathbf{R}_{t-1}$$

## B.4 $\mathcal{H}$ MATRICES WITH STRONG AND WEAK ADMISSIBILITY

In the recurrent formulation of Log-Linear Attention, although there are $\mathcal{O}(\log T)$ states corresponding to different hierarchical levels, roughly half of them are zero in practice. This sparsity arises from the specific structure of HODLR matrices, which belong to a broader class of $\mathcal{H}$ matrices known as *weakly admissible* Hackbusch et al. (2004).

---

[15]In fact, our initial attempts involved using fully Hierarchical matrices, but we were unable to derive a recurrent formulation with $\mathcal{O}(\log T)$ complexity. This motivated the design of Quasi-Hierarchical matrices specifically to support efficient recurrence.

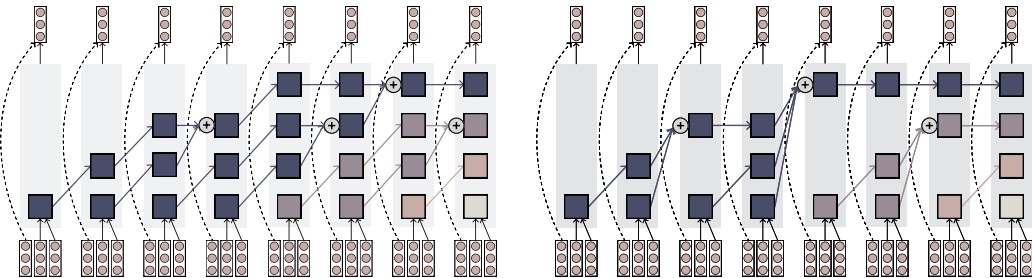

**Figure 7: Left**: $\mathcal{H}$ matrices with strong admissibility. **Right**: $\mathcal{H}$ matrices with weak admissibility.

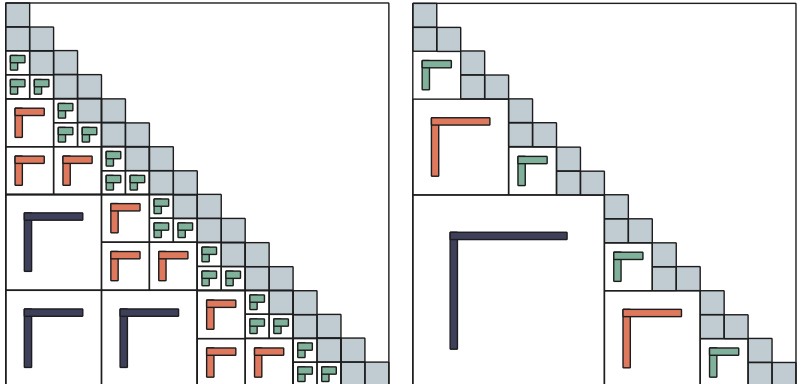

**Figure 8: Left**: $\mathcal{H}$ matrices with strong admissibility. **Right**: $\mathcal{H}$ matrices with weak admissibility.

Figures 8 and 7 illustrate an alternative structure based on strong (or standard) admissibility. Unlike the weakly admissible variant, strongly admissible $\mathcal{H}$ matrices allow for finer-grained partitioning of the matrix, and their corresponding recurrent forms utilize all hierarchical levels.

While strong admissibility can yield more accurate approximations, it comes with a significant computational cost Hackbusch et al. (2004). In our early experiments, using strong admissibility in a `Triton` implementation resulted in up to a `4x` slowdown, with only marginal improvements in accuracy. As a result, we adopt the weakly admissible structure throughout this work and refer to it simply as the $\mathcal{H}$-matrix.

## C  IMPLEMENTATIONS

```python
import torch
import numpy as np
import torch.nn.functional as F

def segsum(x):
    T = x.size(-1)
    x_cumsum = torch.cumsum(x, dim=-1)
    x_segsum = x_cumsum[..., :, None] - x_cumsum[..., None, :]
    mask = torch.tril(torch.ones(T, T, device=x.device, dtype=bool))
    x_segsum = x_segsum.masked_fill(~mask, -torch.inf)
    return x_segsum

def level_mask(level, T):
    if level == 0:
        return torch.eye(T, dtype=torch.bool)

    i, j = torch.meshgrid(torch.arange(T), torch.arange(T), indexing="ij")
    half = 1 << (level - 1)
    clipped = i - (i % (1 << level - 1))
    valid = (i % (1 << level) >= half) & (j + half >= clipped) & (j < clipped)
    return valid

def construct_H_matrix(a, L):
    T = a.size(-1)
    A = torch.exp(segsum(a))
    return sum([A * L[..., level, :].unsqueeze(-1) * level_mask(level, T) for level in range(int(np.log2(T)) +
        1)])

def hattention(X, A, B, C, L, block_len=8):
```

```python
33      """
34      Arguments:
35      X: (batch, length, n_heads, d_head)
36      A: (batch, length, n_heads)
37      B: (batch, length, n_heads, d_state)
38      C: (batch, length, n_heads, d_state)
39      L: (batch, length, n_heads, num_levels) where num_levels = log2(length) + 1
40      Return:
41      Y: (batch, length, n_heads, d_head)
42      """
43      T = X.shape[1]
44      assert X.dtype == A.dtype == B.dtype == C.dtype
45      assert X.shape[1] % block_len == 0
46      input_shape = X.shape
47      # Rearrange into blocks/chunks
48      b, cl = X.shape[0], X.shape[1]
49      c = cl // block_len
50      X, A, B, C, L = [x.reshape(b, c, block_len, *x.shape[2:]) for x in (X, A, B, C, L)]
51      A = A.permute(0, 3, 1, 2)  # (batch, n_heads, c, block_len)
52      A_cumsum = torch.cumsum(A, dim=-1)  # (batch, n_heads, c, block_len)
53
54      num_intra_chunk_levels = int(np.log2(block_len)) + 1
55      num_inter_chunk_levels = int(np.log2(T)) + 1 - num_intra_chunk_levels
56      # Partition the lambda into intra-chunk and inter-chunk lambda
57      L_intra, L_inter = L[..., :num_intra_chunk_levels], L[..., num_intra_chunk_levels:]
58      L_intra = L_intra.permute(0, 3, 1, 4, 2)  # (batch, n_heads, num_chunks, num_levels, block_len)
59
60      # Intra-chunk Computation
61      H = construct_H_matrix(A, L_intra)  # Materialize the H matrix as a dense matrix
62      Y_diag = torch.einsum("bclhn,bcshn,bhcls,bcshp->bclhp", C, B, H, X)
63
64      # Inter-chunk Computation
65      decay_states = torch.exp((A_cumsum[..., -1:] - A_cumsum))
66      states = torch.einsum("bclhn,bhcl,bclhp->bchpn", B, decay_states, X)
67      decay_chunk = F.pad(torch.exp(segsum(A_cumsum[..., -1])), (0, 0, 1, 0))[..., :-1, :]
68      state_decay_out = torch.exp(A_cumsum)
69
70      def compute_Y_off_level(states, level):
71          mask = level_mask(level + 1, c).unsqueeze(0).unsqueeze(0)
72          decay_chunk_level = decay_chunk * mask
73          states = torch.einsum("bhzc,bchpn->bzhpn", decay_chunk_level, states)
74          Y_off = torch.einsum(
75              "bclhn,bchpn,bhcl,bclh->bclhp",
76              C,
77              states,
78              state_decay_out,
79              L_inter[..., level],
80          )
81          return Y_off
82
83      Y_off = torch.zeros_like(Y_diag)
84      for i in range(num_inter_chunk_levels):
85          Y_off += compute_Y_off_level(states, i)
86
87      Y = (Y_off + Y_diag).reshape(input_shape)
88      return Y
```

---

**Algorithm 1** Chunkwise Log-Linear Attention Algorithm

---

1: **for** $t \in [T/C]$ **do**
2:     $\mathbf{Y}_{[t]} = \left( \mathbf{Q}_{[t]} \mathbf{K}_{[t]}^\top \odot \mathbf{M}_{[t]}^{\mathcal{H}} \right) \mathbf{V}_{[t]}$
3: **end for**
4:
5: **for** $\ell \in [\log_2 (T/C)]$ **do**
6:     **for** $t \in [T/C]$ **do**
7:         $\mathbf{Y}_{[t]} = \mathbf{Y}_{[t]} + \text{mask}_{\mathbf{Q}}^{(\ell)} \left( \mathbf{\Lambda}_{[t]}^{(\ell)} \odot \mathbf{Q}_{[t]} \mathbf{S}_{[t]} \right)$
8:         $\mathbf{S}_{[t+1]} = \text{mask}_{\mathbf{A}}^{(\ell)} \left( \mathbf{A}_{[t]} \mathbf{S}_{[t]} \right) + \text{mask}_{\mathbf{K}}^{(\ell)} \left( \mathbf{K}_{[t]} \mathbf{V}_{[t]}^\top \right)$
9:     **end for**
10: **end for**
11: **return** $\mathbf{Y}$

---

A naive implementation computes each level independently using a Mamba-2-style primitive, then sums the outputs—leading to redundant memory access and kernel launches. To improve efficiency, we fuse computation across four levels into a single Triton kernel, which we found optimal given SRAM constraints on an H100.

For backpropagation, we unify gradient computation across all levels for $\nabla\mathbf{K}$ and $\nabla\mathbf{V}$ by analytically factoring their dependencies. This reduces kernel count and improves memory efficiency, achieving over 3× speedup compared to the naive multi-level version.

# D    ADDITIONAL EXPERIMENT DETAILS

For the implementation benchmarks, all experiments were conducted on an H100 GPU with a batch size of 2, using 48 attention heads, a head dimension of 64, and a chunk size of 64. In Mamba-2-style models, the attention heads are applied to $\mathbf{V}$ (MVA pattern), whereas in FlashAttention-2, we adopt GQA-style attention by applying heads to $\mathbf{Q}$. The dimensions of the $\mathbf{Q}$ and $\mathbf{K}$ states are set to 128, aligning with common training configurations.

For the MQAR experiments, we largely follow the setup described in Arora et al. (2024). Models are trained and evaluated on 256-token sequences containing between 4 and 64 key-value pairs. We do not evaluate on sequences longer than those used in training (i.e., no length generalization). In (Log-Linear) Mamba-2 models, both the state and head dimensions are set to 16. For (Log-Linear) Gated DeltaNet, we use two attention heads by default, except for models with a dimension of 16, where a single head is used. We tune the learning rate and, for Log-Linear models, also tune the parameterization of $\lambda$. We run each configuration with five seeds. Training was early stopped when accuracy exceeded 99%.

For the language modeling experiments, each run was performed on $8\times$A100 or $8\times$H100 GPUs over the course of several days. We do not tie word embeddings, use a vocabulary size of 32,000, and set the initializer range to 0.006. Training is performed with a global batch size of approximately 524K tokens for 95K steps (roughly 50B tokens). We use the `flash-linear-attention` and `flame` libraries Yang & Zhang (2024); Zhang & Yang (2025), following most of their default configurations.

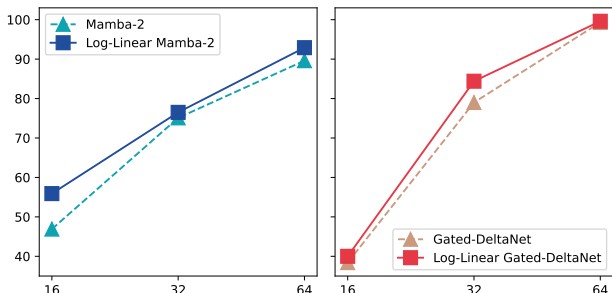

**Figure 9:** MQAR experiments with early stopping at 99% accuracy.

**Detailed Experimental Results.** Figures 9 and 10 and Tables 6 and 7 provide detailed results.

| Model | Wiki. ppl $\downarrow$ | LMB. ppl $\downarrow$ | LMB. acc $\uparrow$ | PIQA acc $\uparrow$ | Hella. acc_n $\uparrow$ | Wino. acc $\uparrow$ | ARC-e acc $\uparrow$ | ARC-c acc_n $\uparrow$ | Avg. |
|---|---|---|---|---|---|---|---|---|---|
| Transformer | 21.56 | 22.14 | 38.8 | 65.1 | 39.6 | 50.7 | 45.6 | 24.5 | 44.0 |
| w/ *24 Layers* | 21.13 | 21.17 | 39.3 | 66.6 | 40.4 | 53.3 | 47.8 | 26.4 | 45.6 |
| Mamba-2 | 22.44 | 24.14 | 36.2 | 66.8 | 41.2 | 51.6 | 46.0 | 27.1 | 44.8 |
| w/ *Log-Linear* | 22.11 | 21.86 | 37.0 | 66.6 | 41.1 | 51.7 | 45.5 | 27.4 | 44.9 |
| Gated DeltaNet | 21.73 | 19.71 | 39.3 | 65.8 | 40.9 | 52.2 | 47.1 | 24.6 | 45.0 |
| w/ *Log-Linear* | 21.44 | 18.08 | 40.5 | 66.1 | 41.4 | 53.9 | 46.9 | 24.9 | 45.6 |

**Table 6:** Performance comparison on language modeling and zero-shot commonsense reasoning.

# E    LLM USAGE

In this work, large language models (LLMs) were used to enhance writing by improving clarity and conciseness, to identify relevant literature across and beyond the immediate domain, and to support research ideation, particularly in mathematics and coding.

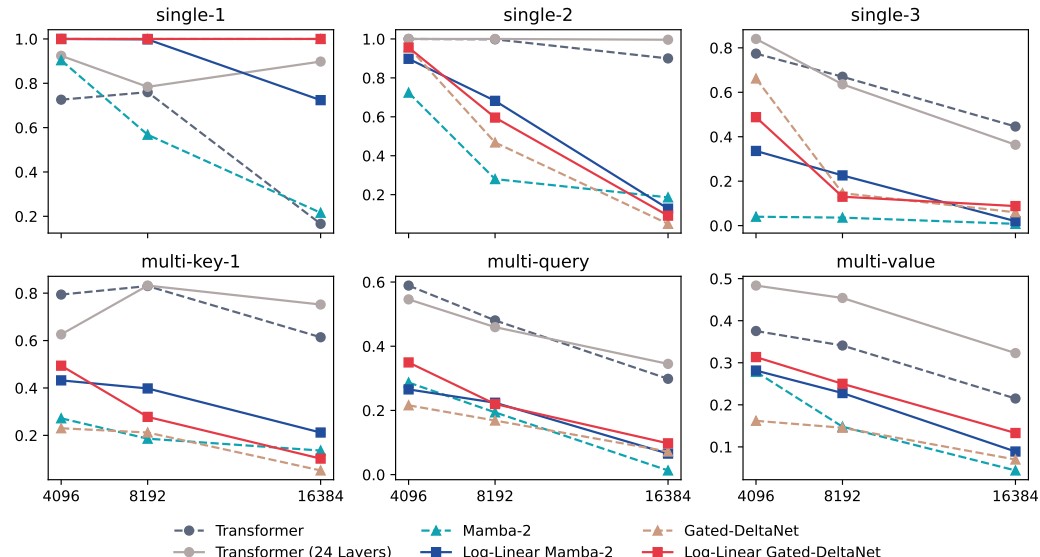

**Figure 10:** Needle-In-A-Haystack experiments. See Table 4 for details.

| Model | SWDE | | | | SQuAD | | | | FDA | | | |
|---|---|---|---|---|---|---|---|---|---|---|---|---|
| | 512 | 1024 | 2048 | 16k | 512 | 1024 | 2048 | 16k | 512 | 1024 | 2048 | 16k |
| Transformer | 47.3 | 44.6 | 45.2 | 45.4 | 34.0 | 34.5 | 34.5 | 34.5 | 72.2 | 70.8 | 72.9 | 72.2 |
| w/ *24 Layers* | 53.8 | 50.9 | 50.3 | 50.8 | 30.7 | 31.2 | 31.2 | 30.9 | 73.8 | 76.0 | 74.4 | 73.8 |
| Mamba-2 | 42.5 | 37.7 | 30.7 | 30.6 | 21.6 | 21.7 | 21.9 | 22.0 | 53.7 | 38.0 | 23.8 | 21.3 |
| w/ *Log-Linear* | 41.9 | 35.6 | 28.4 | 28.5 | 25.8 | 25.9 | 25.9 | 26.1 | 53.0 | 37.5 | 20.5 | 16.6 |
| Gated DeltaNet | 41.0 | 32.5 | 27.2 | 27.8 | 23.8 | 24.1 | 24.3 | 23.7 | 57.2 | 43.7 | 33.2 | 30.5 |
| w/ *Log-Linear* | 46.2 | 39.4 | 35.3 | 35.1 | 25.2 | 25.2 | 25.3 | 25.3 | 64.9 | 53.5 | 39.1 | 30.5 |

| Model | TriviaQA | | | | Drop | | | | NQ | | |
|---|---|---|---|---|---|---|---|---|---|---|---|
| | 512 | 1024 | 2048 | 16k | 512 | 1024 | 2048 | 16k | 512 | 1024 | 2048 |
| Transformer | 48.5 | 49.6 | 48.5 | 48.5 | 22.8 | 22.8 | 22.5 | 22.3 | 24.5 | 24.3 | 24.6 |
| w/ *24 Layers* | 46.9 | 47.0 | 46.8 | 46.8 | 22.7 | 22.4 | 22.7 | 23.0 | 24.0 | 24.4 | 24.5 |
| Mamba-2 | 43.7 | 43.2 | 43.2 | 43.2 | 22.2 | 22.1 | 22.2 | 22.1 | 18.5 | 16.5 | 16.5 |
| w/ *Log-Linear* | 44.9 | 45.0 | 45.5 | 45.5 | 20.2 | 20.6 | 20.3 | 19.9 | 20.0 | 19.9 | 20.4 |
| Gated DeltaNet | 45.6 | 45.6 | 45.6 | 45.6 | 21.1 | 21.7 | 21.4 | 21.8 | 20.1 | 18.4 | 18.7 |
| w/ *Log-Linear* | 45.9 | 45.6 | 46.0 | 46.0 | 20.7 | 20.8 | 20.8 | 21.0 | 22.5 | 21.8 | 21.3 |

**Table 7:** Accuracy on retrieval tasks w/ input truncated to different lengths.

| Model | Single-Doc QA | | | Multi-Doc QA | | | Summarization | | | Few-shot | | | Code | |
|---|---|---|---|---|---|---|---|---|---|---|---|---|---|---|
| | NQA | QQA | MFQ | HQA | 2WM | Mus | GvR | QMS | MNs | TRC | TQA | SSM | LCC | RBP |
| Transformer | 11.7 | 9.7 | 20.8 | 22.4 | 29.8 | 6.7 | 13.1 | 9.4 | 3.2 | 27.5 | 28.0 | 16.2 | 23.7 | 29.8 |
| w/ *24 Layers* | 10.7 | 18.4 | 26.1 | 33.7 | 25.7 | 11.6 | 16.8 | 9.4 | 10.3 | 16.5 | 45.2 | 14.3 | 31.5 | 30.9 |
| Mamba-2 | 9.1 | 17.4 | 10.9 | 11.2 | 20.9 | 4.3 | 8.3 | 6.0 | 4.9 | 2.0 | 22.6 | 8.8 | 38.1 | 34.6 |
| w/ *Log-Linear* | 9.8 | 9.6 | 15.4 | 11.5 | 22.0 | 5.1 | 5.4 | 11.1 | 4.5 | 16.5 | 21.6 | 14.9 | 31.2 | 30.3 |
| Gated DeltaNet | 8.5 | 11.9 | 16.4 | 14.4 | 24.5 | 6.6 | 9.2 | 11.7 | 11.6 | 36.5 | 25.3 | 23.1 | 31.1 | 31.1 |
| w/ *Log-Linear* | 9.9 | 6.1 | 17.6 | 17.7 | 25.2 | 7.5 | 5.5 | 11.9 | 1.9 | 8.0 | 41.1 | 23.2 | 28.3 | 29.6 |

**Table 8:** Accuracy on LongBench tasks (Bai et al., 2023): Narrative QA, QasperQA, MultiField QA, HotpotQA, 2WikiMultiQA, Musique, GovReport, QMSum, MultiNews, TREC, TriviaQA, SamSum, LCC, and RepoBench-P.

