# OpenReview forum: "Log-Linear Attention"
_ICLR.cc/2026/Conference — ICLR 2026 Poster_

### Official Review · Reviewer_YX7t · 2025-10-28

**Soundness:** 3
**Presentation:** 3
**Contribution:** 3
**Rating:** 4
**Confidence:** 4

**Summary:**

Log-Linear Attention is an extension of Linear Attention to multiple timescales growing logarithmically along with a Fenwick-tree scheme. This way it introduces more memory capacity reserved for short time scales, which in turn enables a clearer separation of the longer-term memory (higher levels) for long-scale information. Through input-dependent temporal coefficients this time-scale separation can be achieved. The method shows slight performance gains on synthetic long-context and memory-capacity tasks.

**Strengths:**

- fast parallel and hardware-aware implementation
- captures typical inductive bias on shorter time-scales (recency bias)

**Weaknesses:**

- theoretically unclear why the extended memory can be effectively used, except for not "bloating" the long-term memory at the highest level with short time-scale information that can be store in lower levels
- mild improvements on benchmarks
- unclear scaling behavior

**Questions:**

- How does your method relate to other existing hierarchical sequence architectures like WaveNet that uses dilated convolutions [1]?
- How would this combine with compression / focusing of the inputs via exponential gating as in xLSTM [2] which has shown to be beneficial for long-context tasks in [3]?
- Given the different temporal scales are only separated by the different temporal coefficients $\lambda_t^{(l)}$, how can the effective network capacity exceed the one of pure linear attention based on the foundational theory of Hopfield network capacity? For a querying mechanism that should work across all temporal scales (as in MQAR), shouldn't the "space benefit" vanish to normal linear attention (potentially there is less noise on the shorter time-scales in Log-Linear Attention)?


[1] van den Oord et al. (2016): WaveNet: A generative model for raw audio

[2] Beck et al. (2024): xLSTM: Extended Long Short-Term Memory

[3] Beck et al. (2025): xLSTM 7B: A Recurrent LLM for Fast and Efficient Inference

---

> ### Author Response · Authors · 2025-11-18
> **Response**
>
> Thank you for the thoughtful comments and suggestions!
>
> ---
>
> ## Theoretical Explanation (W1)
>
> Great question. Intuitively, if $\\lambda^{(\\ell)}\_t = 1$ for all $t$ and all $\\ell$, then log-linear attention becomes exactly equivalent to standard linear attention or SSMs.
>
> More precisely, an equivalence holds whenever $\\lambda^{(\\ell)}\_t$ and $\\lambda^{(\\ell^{\\prime})}\_t$ are linearly related over time. Many linear-attention mechanisms and SSMs mix sequences through structured matrix multiplication using  semi-separable matrices (SSS) [1]. Log-linear attention performs this mixing via the $\\mathcal{H}$ matrix. Under certain conditions, $\\mathcal{H}$ matrices collapse to hierarchically semi-separable matrices (HSS) [2].
>
> [1] https://arxiv.org/abs/2405.21060
> [2] https://arxiv.org/abs/1909.07909
>
> ---
>
> ## Mild Improvements (W2)
>
> We acknowledge that our improvements over baselines are small (as noted in the paper). In our view, the primary contribution of this work is conceptual rather than empirical. In particular, we show that there is an interesting modeling space that has not been previously explored, i.e., a family of architectures that can serve as a middle ground between linear- and softmax-attention. We hope that this will inspire more future work in this space.
>
> ---
>
> ## Scaling Behavior (W3)
>
> Thank you for the suggestion. Due to constraints typical in academic research, we lack the resources to conduct extensive scaling experiments. Nevertheless, we conducted additional experiments by varying the hidden size (and corresponding head dimension) in the Log-Linear Gated DeltaNet model. The results are presented below.
>
>
> | Hidden Size | WikiText PPL |
> |-------------|--------------|
> | 256         | 92.4         |
> | 512         | 55.9         |
> | 1024        | 40.5         |
>
> ---
>
> ## Connections to Dilated Convolution (Q1)
>
> There is a loose conceptual analogy. Log-linear attention is a hierarchical extension of the mixing step used in some linear attention and SSM mechanisms. Dilated convolution is a hierarchical extension of convolution itself. Log-linear attention expands the sequence mixing within each layer, whereas dilated convolution expands the receptive field across layers.
>
> From an algebraic viewpoint, non-dilated convolution corresponds to structured matrix multiplication by a Toeplitz matrix. Non-log-linear attention corresponds instead to a sequentially semi-separable (SSS) matrix. Toeplitz matrices lie within the broader family of low-displacement rank matrices, while SSS matrices belong to the class of semi-separable matrices. To our knowledge, the two classes differ, though matrices can be engineered to satisfy both structural constraints.
>
> [Various Categories of Structured Matrices, screenshot from [3]](https://ibb.co/RGqQtCqX)
> [3] https://link.springer.com/article/10.1007/s00354-023-00226-1
>
> ---
> ## Toward Log Linear xLSTM? (Q2)
>
> This is an interesting direction! In principle, constructing a log-linear variant of xLSTM (particularly mLSTM) appears possible, but a practical realization would require substantial kernel-level modifications.
>
> Log-linear attention begins by noting that parallel sequence mixing can be expressed as structured matrix multiplication by a sequentially semi-separable matrix. Elevating this structure across levels produces an $\\mathcal{H}$ matrix. To see how an xLSTM-style (notably, mLSTM) update might fit into this framework, consider the recurrence
>
> $$
> \\mathbf{S}\_t \= f\_t \\mathbf{S}\_{t-1} + i\_t \\mathbf{v}\_t \\mathbf{k}\_t^{\\top},
> \\quad
> \\mathbf{z}\_t \= f\_t \\mathbf{z}\_{t-1} + i\_t \\mathbf{k}\_t,
> \quad
> \\mathbf{o}\_t \= \\frac{\\mathbf{S}\_t \mathbf{q}\_t}{\max \left(1, \lvert \\mathbf{z}\_t^{\\top} \\mathbf{q}\_t \rvert\\right)}
> $$
>
> To simplify the argument, suppose $\\mathbf{z}\_t$ is given. One can then show that the output satisfies
> $
> \\mathbf{O} = \\left(\\mathbf{Q}\\mathbf{K}^{\\top} \\odot \\mathbf{L} \\right)\\mathbf{V},
> $
> for some semiseparable mask $\\mathbf{L}$. This representation suggests a route toward a log-linear mLSTM. Operationally, such a model would maintain on the order of $\mathcal{O}(\log T)$ states. This remains speculative but appears promising!
>
> ---
> ## Hopfield Networks and Associative Memory (Q3)
>
> Since Gated DeltaNet admits an interpretation as test time regression on an associative recall objective, log-linear attention does not fundamentally change this view beyond increasing effective state size.
>
> Similarly, one could plausibly construct a log-linear MesaNet [4] (the arguments will be slightly more involved, but I'm happy to elaborate if you are interested). Because MesaNet already optimizes an associative recall objective, the main effect of log linearity would primarily be a larger or more flexible state space, not a change to the underlying retrieval objective.
>
> This perspective may help explain why log-linear attention does not dramatically outperform DeltaNet on MQAR.
>
> [4] https://arxiv.org/abs/2506.05233

---

> > ### Comment · Reviewer_YX7t · 2025-11-24
> >
> > Thanks for all your clarifications! It would be great to add these into your paper, especially the connection to dilated convolution (like WaveNet) and other hierarchical models and how your framework could be extended to other linear RNNs like xLSTM.
> >
> > I'm still a bit skeptical about the improved memory capacity for tasks that should work across all time-scales like MQAR, so I'm happy to hear more about the connection to MesaNet and how you would expect this to be improved.
> >
> > In general, I see this work as a very interesting conceptual direction in between constant-state RNNs and linear-scaling-state Transformers!

---

> ### Author Response · Authors · 2025-11-24
> **Follow-up**
>
> Thanks for the encouraging feedback!
>
> We have updated the paper to incorporate your suggestions (highlighted in red, pages 9 and 10). Does the revised version look better?
>
> On that note, if this answers your question, we would be grateful if you could update the evaluation. And of course, please feel free to let us know if there is anything else you would like us to clarify!
>
> ---
>
> ## Towards Log Linear MesaNet
>
> To derive a log-linear variant of MesaNet, we follow two main steps.
> 1. We note that MesaNet can also be written as a sequence mixing operation with a semi-separable matrix.
> 2. Building on this observation, we introduce $\\mathcal{O}(\\log T)$ distinct temporal coefficients.
>
> The argument is somewhat more involved because, to the best of my knowledge, we can no longer express $\\mathbf{O} \= (\\mathbf{Q} \\mathbf{K}^\\top \\odot \\mathbf{L}) \\mathbf{V}$ for a semi-separable mask $\\mathbf{L}$. Instead, we show that $\\mathbf{O} \= \\mathbf{P} \\mathbf{V}$, where $\\mathbf{P}\_{t,s} \= \\mathbf{Q}\_t \\left( \\mathbf{M}\_{t,s} \\right) \\mathbf{K}\_s^\\top$ and
> $\\mathbf{M}\_{t,s} \= \\mathbf{C}\_t \\left( \\prod\_{i\=t}^{s+1} \\mathbf{A}\_i \\right) \\mathbf{B}\_{s} \\in \\mathbb{R}^{d \\times d}$.
> This definition makes $\\mathbf{M} \\in \\mathbb{R}^{T \\times T \\times d \\times d}$ a four dimensional tensor, or equivalently a two dimensional block matrix. Importantly, $\\mathbf{M}$ exhibits semi-separable-like behavior when viewed across temporal slices.
>
> Appendix A provides additional details on why this lifted representation generalizes Mamba-2, Gated DeltaNet, and (as discussed in this response, new) MesaNet. Using the same technique described there, right before Appendix B, we can add temporal coefficients, possibly matrix-valued, to obtain a log-linear MesaNet. Implementing this construction in practice is beyond the scope of this response.
>
> We include the full derivation below as an image so that it does not make this comment unnecessarily long.
>
> [Figure: Derivation](https://ibb.co/qYy8CZg1)

---

### Official Review · Reviewer_Zys3 · 2025-10-30

**Soundness:** 4
**Presentation:** 4
**Contribution:** 3
**Rating:** 8
**Confidence:** 4

**Summary:**

This paper identifies the fixed state size of currently best performing linear attention variants with gating, such as Mamba-2 and Gated DeltaNet as the main limitation to handle information in a long context.
By introducing log-linear attention - a framework with a logarithmically growing set of hidden states that can be applied to existing linear attention variants - it provides a middle ground between standard attention with linear growing memory and linear attention with a fixed state size, which is independent of sequence length.
Log-linear attention is based on the insight that efficient attention variants depend on the structure of the structure of the masking matrix in the attention operation, and replaces existing masking structures with a hierarchical one.
With log-linear variants of Mamba-2 and Gated Delta Net, the authors demonstrate the general applicability of the log-linear attention framework.
In their experiments on small scale language modeling setups and synthetic tasks the log-linear variants show mild but consistent improvements over the standard linear RNN variants.
In training throughput and runtime benchmarks the authors demonstrate benefits of the log-linear Mamba-2 variant over Flash-attention 2 at longer context and only small runtime overheads compared to the default Mamba-2 implementation.

**Strengths:**

- Clear motivation
- Well written
- The overview of related variants and the view of efficient attention mechanisms as different parametrizations of structured masking matrices is great. It shows how this naturally results in the idea & implementation for log-linear attention.
- To the best of my knowledge log-linear attention is a novel method for expanding the state size.
- The paper provides simple pure PyTorch implementations and shows experiments with optimized kernels (even though code for these is missing) achieving runtime benefits over existing methods

**Weaknesses:**

- Only small performance improvements over linear counter parts / base methods across several tasks (admitted by authors)
- The authors place log-linear attention as middle ground between standard attention and linear attention in terms of memory state size: Hence I would expect an exemplary calculation of the memory consumption of log-linear attention, standard linear attention and KV-cache for various sequence lengths and reasonable model sizes
- No code provided for Mamba2 and Gated Delta Net log linear attention variants
- The paper would further benefit from more details on the efficient kernel implementations (including code) for Mamba-2


Despite these weaknesses, the paper has a clear motivation, is very well written, contributes new insights on efficient attention variants and their implementation, as well as a novel method for expanding the state size of linear attention variants, which outweighs the weaknesses. Therefore I recommend acceptance.

**Questions:**

- L.392-393: Why does the linear layer on top of the hidden states add 3% additional parameters to Mamba-2 and only 0.4% to Gated Delta Net?
- Description of the P matrix L.236 - 245 would help understandability
- L. 337, 997: Could the authors elaborate on the MVA pattern for Mamba-2 and/or provide references to descriptions of this?
- Does there exist an optimized kernel implementation for the log-linear variant of Gated Delta Net?

---

> ### Author Response · Authors · 2025-11-18
> **Response**
>
> Hi, thanks for the encouraging suggestions and comments!
>
> ## Memory Footprint (W2)
>
> In our response, we will focus primarily on decoding and, in particular, on the number of levels, which represent the linear-attention or SSM states maintained during inference. This emphasis differs slightly from your question on an exemplary calculation, but we highlight this quantity because it is invariant to architectural choices such as head dimension or number of heads and is therefore more intuitive. During training, we simply set the number of levels to the maximum.
>
> During decoding, the peak number of required levels grows approximately logarithmically with sequence length. Interestingly, although the peak can be large, the number of **active** levels fluctuates and averages to only slightly above half of the maximum.
>
> [Figure 1: Number of Levels](https://ibb.co/BHHb26cW)
>
> At first glance, it may seem surprising that only a bit more than half of the levels remain active on average. This behavior follows directly from the Fenwick-Tree partitioning and, in particular, from the sparsity pattern of its underlying $\\mathcal{H}$ matrix. Such patterns are characteristic of **HODLR** matrices, which belong to the broader family of weakly admissible $\mathcal{H}$-matrices [1].
>
> This observation naturally raises the question of whether another structure might be more suitable. We also implemented an alternative based on **strong (standard) admissibility**. Strongly admissible $\\mathcal{H}$-matrices allow finer partitioning, and their associated recurrent forms utilize *all* hierarchical levels.
>
> [Figure 2: Strongly vs. Weakly Admissible $\\mathcal{H}$ Matrices (Appendix B.4)](https://ibb.co/gMP1t3bN)
>
> While strong admissibility can provide marginally more accurate approximations, it introduces substantial compute overhead. Early Triton prototypes showed up to a 4x slowdown, with negligible improvement in loss. Consequently, we adopt weak admissibility throughout.
>
> | | Runtime (8 x H100) | Smoothed Loss (Final Step) |
> | :--- | :--- | :--- |
> | Weak-Admissible $\\mathcal{H}$ (Fenwick Tree) | 1.75 days | 2.12 |
> | Strong-Admissible $\\mathcal{H}$ | 6.25 days | 2.13 |
>
> [1] https://link.springer.com/article/10.1007/s00607-004-0080-4
>
> ---
>
>
> ## Implementation notes (W3)
>
> We include pseudocode in Appendix C, and the full implementation will be released once the anonymity period ends. In the meantime, we provide an example Triton kernel snippet below.
>
> [Example Triton Kernel](https://anonymous.4open.science/r/iclr2025-response-D007/chunk.py)
>
> ---
> ## Kernel Optimization Details (W4)
>
> In a naive implementation of log-linear attention, each of the $\\mathcal{O}(\\log T)$ levels in the Fenwick-partitioned hierarchy is computed independently using a linear attention primitive (e.g., Mamba-2 or Gated DeltaNet). While conceptually straightforward, this level-by-level approach introduces significant redundancy: each invocation launches a separate kernel, leading to repeated memory accesses and KV cache computations.
>
> Our optimization fuses the computation of multiple levels into a single custom kernel. This fused kernel amortizes memory access and KV cache costs across levels, significantly improving efficiency. However, fusing more levels increases register and shared memory usage, which can exceed on-chip SRAM capacity. To balance this trade-off, we empirically tune the number of levels fused per kernel.
>
> In the backward pass, we analytically factor the gradients so that the computation for keys (K) and values (V) is shared across levels, rather than repeated per level. This improves both runtime and memory bandwidth efficiency. This optimization leverages a property of the specific class of HODLR matrices we use, where values within each row of a low-rank block are identical.
>
> Figure 4 compares the optimized implementation with a naive one that repeatedly applies existing Mamba-2 primitives. For convenience, we summarize the runtime performance below:
>
> | Sequence Length | Optimized (ms) | Naive (ms) |
> |-----------------|----------------|------------|
> | 4096            | 6.1            | 19.4       |
> | 8192            | 13.0           | 42.6       |
> | 16384           | 28.5           | 97.1       |
> | 32768           | 63.8           | 232.9      |
> | 65536           | 138.7          | 515.3      |
> | 131072          | 302.5          | 1127.8     |
>
> ---

---

> ### Author Response · Authors · 2025-11-19
> **Response 2/2**
>
> ## Different Overheads (Q1)
>
> Great observation. The overheads depend on many factors, such as the head dimension, the number of heads, and whether multi-value attention (MVA) is enabled (more below). Hence, it is natural to see different overheads across experiments.
>
> ---
>
> ## $\\mathbf{P}$ Matrix (Q2)
>
> Here is a bit of background. Hierarchical matrices ($\\mathcal{H}$ matrices) usually follow a structure similar to the figure below. The diagonal block entries are dense, and the off-diagonal blocks are hierarchically low rank.
>
> [Figure 3: $\\mathcal{H}$ Matrix](https://ibb.co/srmyhK9)
>
> Recall that $\\mathbf{P} = \\mathbf{Q}\\mathbf{K}^{\\top} \\odot \\mathbf{M}^{\\mathcal{H}}$. The term $\\mathbf{Q}\\mathbf{K}^{\\top}$ is a low-rank matrix. The mask $\\mathbf{M}^{\\mathcal{H}}$, induced by the Fenwick tree partitioning, turns $\\mathbf{Q}\\mathbf{K}^{\\top}$ into a hierarchical matrix. The chunkwise algorithm in Section 3.3 uses the fact that  $\\left(\\mathbf{Q}\\mathbf{K}^{\\top} \\odot \\mathbf{M}^{\\mathcal{H}}\\right)\\mathbf{V}$  reduces to a structured matrix multiplication with an $\\mathcal{H}$ matrix, which can be computed in about $\\mathcal{O}(T \\log T)$ time.
>
> Appendix B provides additional detail, too, and let us know if this is clear!
>
> ---
>
> ## Multi-Value Attention (MVA, Q3)
>
> For the full explanation, see Section 7.2 of the Mamba 2 paper [3]. At a high level:
>
> * Multi-query attention (MQA) uses H query heads but only one key and one value head.
> * Multi-value attention (MVA) instead uses one query head and one key head, but H value heads.
>
> Here is a simple summary table based on Equation 20 of Mamba 2 (written in standard attention notation):
>
> | Variant        | Q | K | V |
> |----------------|---|---|---|
> | Multi-head     | H | H | H |
> | Multi-query    | H | 1 | 1 |
> | Multi-key      | 1 | H | 1 |
> | Multi-value    | 1 | 1 | H |
>
> [2] https://arxiv.org/pdf/2405.21060
>
> ---
>
> ## Log-Linear Gated DeltaNet Implementations (Q4)
>
> We also have a Triton implementation of Log-Linear Gated DeltaNet. It follows the same high-level structure as Log-Linear Mamba 2 but the implementation is a bit more involved. Some operations are still unfused and handled by PyTorch because we reused several existing Gated DeltaNet Triton ops. Even so, it is good enough for the models we train. If you would like more details, we are happy to elaborate on them!
>
> ---

---

### Official Review · Reviewer_xdQ7 · 2025-11-01

**Soundness:** 4
**Presentation:** 3
**Contribution:** 4
**Rating:** 8
**Confidence:** 4

**Summary:**

This work proposes a method to improve the modeling performance of modern SSMs/ Linear RNNs such as Mamba and Gated DeltaNet, whose recurrent state can be formulated as $S_{t} = \alpha_t * S_{t-1} + (KV)_t$. To get final output, these algorithms multiply query $Q_t$ by a single state matrix $S_t$, which aggregates information about all previous KV states up to step $t$. Log-linear attention instead disaggregates $S_t$ into $O(\log T)$ states of the same size, each containing information about a disjoint contiguous subsequence of  $\{0, …, t\}$. Then the query gets multiplied independently by each state and also by an input-dependent scalar coefficient $\lambda_i, \; i \in \{0, …, O(\log T)\}$, corresponding to that state. Finally, the results sum up. A pure linear attention variant is equivalent to log-linear attention where $\lambda_i=1$ for all i.

The partitioning of the sequence proceeds according to Fenwick tree scheme, where each subsequence can have at most $2^i$ consecutive timesteps, and the shortest subsequences is located at the latest step t, while longer subsequences’s boundaries go toward the sequence start.

I feel excited about this novel method (see strengths) and vote for its acceptance.

**Strengths:**

* Potent sub-quadratic runtime alternatives to Transformers is an important open area of research, and this work provides a promising way to improve modeling quality of such architectures, hopefully bringing us closer to an algorithm capable of fully replacing Transformer in autoregressive language modeling.

* The proposed method is coherent and intuitive: if we want to increase long-range performance in comparison with pure linear-time algorithms such as Mamba and DeltaNet, it is plausible that we have to execute a higher relative amount of computations, than for short sequences.

* Log-linear attention is a meta-algorithm, which is compatible with many linear-time alternatives to softmax attention.

* I believe the proposed approach has a potential to be extended and generalized by using other partitioning schemes which could bring further performance gains.

* I’d like to specifically praise comprehensive and fair comparisons which don’t shy away from presenting both positive and negative results. They help to build an honest and complete picture of the method’s strong and weak sides and possible areas of application.

* The validation shows meaningful improvements relative to pure linear counterparts on a large subset of tested benchmarks, effectuated by log-linear extension.

**Weaknesses:**

My judgement is that the paper doesn't have major problems. There are some minor issues mostly related to exposition/ formatting which I listed below.

1. Did you perform any measurements of the memory footprint of the algorithm during inference (prefill, decode) and training workloads? A comparison for different sequence lengths with vanilla Mamba-2 and Gated DeltaNet, as well as with FlashAttention would be helpful. I understand that it’s likely to be $O(\log(T))$ times greater than aforementioned algorithms, but that’s an expected trade-off, which is easily tolerated given log-linear attention’s superior modeling quality. Nonetheless, these numbers would be important for finding out the most fitting conditions to use the algo.

2. It’s not clear to me from the paper how the log-linear part of the algorithm modifies the underlying linear part, both in chunk-wise parallel and recurrent algos. For example, what happens when two states for neighboring filled buckets get merged after a recurrent step? Since the states are created independently, $S_{[t_1:t_2)} + S_{[t_2:t_3)} \neq S_{[t_1:t_3)}$, although the underlying algorithm requires precisely $S_{[t_1:t_3)}$. Similar ambiguities emerge when considering chunk-wise form.

3. There is no formal definition of the exact functional form of $\lambda_i$s. Are they simply linear projections of input vector query $q_t$? Or are they calculated using the same laws as alphas (i.e. in a different manner for each underlying architecture)? How does the algorithm handle that the number of lambdas $max(i)$ is not bounded from above and can extrapolate beyond the maximal value during pre-training?

4. Minor typos/ formatting problems:
* Line 189 – why is the right bound $t$ open? From the text it follows that the t-th token itself is a part of the partition.

* Line 213 – it would be clearer to mention explicitly that $b_t^{(i)}$ denotes the starting position of partition i, it was somewhat hard to infer for me at first glance.

* Lines 237-246 – there’s no caption of this figure.

* Line 291 – I believe it’s $\lceil T/C \rceil$.

* Line 763 – image and table captions are overlaid.

* Table 3 – There’s no mention of the table in the text, and it takes an attentive reader to understand that it’s the summary of Table 6.

**Questions:**

1. Can you come up with a theoretical explanation why log-linear attention offers performance improvements in comparison with pure linear variants?

2. Did you try any other partition schemes besides Fenwick Tree partitioning? I could think of other schemes, with overlapping and disjoined partitions. There could even be some schemes that recover $O(T)$ complexity (e.g., proceed as usual until sequence length is X, then keep placing the oldest tokens into the outermost bucket instead of creating new levels of hierarchy). As such, why did you choose this specific scheme?

3. A follow-up question: could there be some trade-off, where this algorithm could run in linear time at the expense of an arbitrary higher memory consumption?

---

> ### Author Response · Authors · 2025-11-17
> **Response**
>
> Hi, thanks for the thoughtful questions and suggestions!
>
> ## Memory Footprint (W1)
>
> In our response, we will focus primarily on decoding and, in particular, on the number of **levels**, the number of linear-attention or SSM states maintained during inference. We emphasize this quantity because it is invariant to architectural choices such as head dimension or number of heads. During training, we simply fix the number of levels to the maximum.
>
> During decoding, the peak number of required levels grows approximately logarithmically with sequence length. Interestingly, although the peak can be large, the number of **active** levels fluctuates and averages to only slightly above half of the maximum.
>
> [Figure 1: Number of Levels](https://ibb.co/BHHb26cW)
>
> This gap between peak and active counts occurs because of **state merging** (discussed below). Merging can zero out certain states, allowing them to be discarded. This phenomenon is also closely tied to your question about the Fenwick Tree representation, which we address next.
>
> ---
>
> ## Fenwick Tree Partitioning (Q2)
>
> At first glance, it may seem surprising that only a bit more than half of the levels remain active on average. This behavior follows directly from the Fenwick-Tree partitioning and, in particular, from the sparsity pattern of its underlying $\\mathcal{H}$ matrix. Such patterns are characteristic of **HODLR** matrices, which belong to the broader family of weakly admissible $\mathcal{H}$-matrices [1].
>
> We also implemented an alternative based on **strong (standard) admissibility**. Strongly admissible $\\mathcal{H}$-matrices allow finer partitioning, and their associated recurrent forms utilize *all* hierarchical levels.
>
> [Figure 2: Strongly vs. Weakly Admissible $\\mathcal{H}$ Matrices (Appendix B.4)](https://ibb.co/gMP1t3bN)
>
> While strong admissibility can provide marginally more accurate approximations, it introduces substantial compute overhead. Early Triton prototypes showed up to a 4x slowdown, with negligible improvement in loss. Consequently, we adopt weak admissibility throughout.
>
> | | Runtime (8 x H100) | Smoothed Loss (Final Step) |
> | :--- | :--- | :--- |
> | Weak-Admissible $\\mathcal{H}$ (Fenwick Tree) | 1.75 days | 2.12 |
> | Strong-Admissible $\\mathcal{H}$ | 6.25 days | 2.13 |
>
> [1] https://link.springer.com/article/10.1007/s00607-004-0080-4
>
> ---
>
> ## State Merging (W2)
>
> This is an excellent question and the mechanism is subtle but important.
>
> Each state is indexed by both time $t$ and level $\\ell$. Your expression $\\mathbf{S}\_{[t\_0:t\_1]}$ is more precisely written as $\\mathbf{S}^{[t\_0:t\_1]}\_{t}$, which we abbreviate as $\\mathbf{S}^{(\\ell\_0)}\_{t}$.
>
> Suppose at time $t$ we maintain two states at levels $\\ell\_0$ and $\\ell\_1$ with coefficients $\\lambda^{(\\ell\_0)}\_t$ and $\\lambda^{(\\ell\_1)}\_t$. The output contribution (Equation 3) is
>
> $$
> \lambda^{(\ell\_0)}\_t \mathbf{q}\_t^\top \mathbf{S}^{(\ell\_0)}\_t
> +
> \lambda^{(\ell\_1)}\_t \mathbf{q}\_t^\top \mathbf{S}^{(\ell\_1)}\_t.
> $$
>
> At a later time $t^\\prime$ these levels may merge into a single level $\\ell\_2$, now sharing a common coefficient $\\lambda^{(\\ell\_2)}\_{t^\\prime}$:
>
> $$
> \lambda^{(\ell\_2)}\_{t^\prime} \mathbf{q}\_{t^\prime}^\top \mathbf{S}^{(\ell\_0)}\_{t^\prime}
> +
> \lambda^{(\ell\_2)}\_{t^\prime} \mathbf{q}\_{t^\prime}^\top \mathbf{S}^{(\ell\_1)}\_{t^\prime}
> \=
> \lambda^{(\ell\_2)}\_{t^\prime} \mathbf{q}\_{t^\prime}^\top
> \left(
> \mathbf{S}^{(\ell\_0)}\_{t^\prime} +
> \mathbf{S}^{(\ell\_1)}\_{t^\prime}
> \right).
> $$
>
> Thus the two states can be merged:
>
> $$
> \mathbf{S}^{(\ell\_2)}\_{t^\prime}
> \=
> \mathbf{S}^{(\ell\_0)}\_{t^\prime}
> +
> \mathbf{S}^{(\ell\_1)}\_{t^\prime}.
> $$
>
> This omits gating and several details, but it captures the central algebraic idea behind state merging.
>
> ---
>
> ## Lambda Parameterization (W3, part 1)
>
> Our $\\lambda$ parameterization follows the gating formulation introduced in Mamba-2. The coefficients are learned functions of token-wise inputs. We explored several parameterization ranges, summarized below:
>
>
> | Range              | Smoothed Loss (Mid-Training) |
> |--------------------|-------------------------------|
> | $(-\\infty, +\\infty)$ | 2.41 |
> | $[0, +\\infty)$       | 2.40 |
> | $[0, 1]$             | 2.42 |
>
> For language modeling, we use the $[0, +\\infty)$ range. For MQAR tasks, this is further tuned.
>
> A representative form for $[0, +\\infty)$ case is
>
> $$
> \lambda\_t^{(\ell)} = \operatorname{softplus}\\!\left( \mathbf{x}\_t \mathbf{W}\_{\lambda}^{(\ell)} \lambda^{(\ell)} \right)
> $$
>
> where $\\mathbf{x}\_t$ is the input vector and both $\\mathbf{W}\_{\\lambda}^{(\\ell)}$ and $\\lambda^{(\\ell)}$ are learnable parameters.

---

> ### Author Response · Authors · 2025-11-18
> **Response (2/2)**
>
> ## Lambda Parameterization and Extrapolation (W3, part 2)
>
> Extrapolation is indeed a current limitation, not of the log-linear attention mechanism itself, but of the particular parameterization of $\\lambda$ that we currently use.
>
> However, it is not inherently infeasible to use a parameterization that, in principle, supports extrapolation. In early experiments, we explored the alternative formulation
>
> $$
> \\lambda^{(\\ell)}\_t
> \=
> \\mathrm{softplus}\\!\\left(
> \\widetilde{\\lambda}^{(\\ell)}\_t
> \\right)
> $$
>
> $$
> \\widetilde{\\lambda}^{(\\ell)}\_t
> \=
> \\left(
> \\mathrm{rope}(\\mathbf{x}\_t \\mathbf{W}\_0, \\, \\ell)
> \\,\\mathbf{W}\_1
> \\right)
> \\;\\odot\\;
> \\left(
> \\mathrm{rope}(\\lambda, \\, \\ell)
> \\,\\mathbf{W}\_1^{\\prime}
> \\right)
> $$
>
> This variant performed a tiny bit better in early experiments. We did not adopt it in the main work because it incurred a higher computational cost.
>
> ---
>
> ## Theoretical Explanation (Q1)
>
> Great question. Intuitively, if $\\lambda^{(\\ell)}\_t = 1$ for all $t$ and all $\\ell$, then log-linear attention becomes exactly equivalent to standard linear attention or SSMs.
>
> More precisely, an equivalence holds whenever $\\lambda^{(\\ell)}\_t$ and $\\lambda^{(\\ell^{\\prime})}\_t$ are linearly related over time. Many linear-attention mechanisms and SSMs mix sequences through structured matrix multiplication using (sequentially) semi-separable matrices (SSS) [2]. Log-linear attention performs this mixing via the $\\mathcal{H}$ matrix. Under certain conditions, $\\mathcal{H}$ matrices collapse to hierarchically semi-separable matrices (HSS) [3].
>
> [2] https://arxiv.org/abs/2405.21060
> [3] https://arxiv.org/abs/1909.07909
>
> ---
>
> ## Algorithmic Trade-offs (Q3)
>
> This is a great question. As far as I'm aware, no algorithm computes this procedure in strictly linear time. That said, the practical design space is governed by a trade-off between the number of **steps** and the amount of **memory** required.
>
> In Equation 5 of Appendix C, we show that the algorithm can be implemented as running
> $
> \\mathcal{O}\\!\\left(
> \\log \\frac{T}{C}
> \\right)
> $
> number of state passing, where $C$ is the chunk size. When the state passing happens in sequence (in a loop), this implementation has peak on-chip (SRAM) usage comparable to analogous linear-attention approaches.
>
> It is also possible to execute a single state-passing procedure across *all* levels simultaneously, which increases parallelism (fewer steps) at the cost of higher SRAM usage. More generally, we can interpolate between these extremes by chunking along the **level** dimension.
>
> Below is a summary of the complexity for several design choices (using the same simplifications as in Mamba-2, for example, setting state and head dimensions to be equal).
>
> **Compute complexity:**
>
> $$
> O\\left(
> T C N
> \\,+\\,
> \\log\\left(\\frac{T}{C}\\right) T N^{2}
> \\right)
> $$
>
> | **Steps** | **Memory** |
> |-----------|------------|
> | $O\\left(\\frac{T}{C} \\log \\frac{T}{C}\\right)$ | $O\\left(TN + \\left(\\frac{T}{C}\\right) N^{2}\\right)$ |
> | $O\\left(\\frac{T}{C}\\right)$ | $O\\left(TN + \\left(\\frac{T}{C} \\log \\frac{T}{C}\\right) N^{2}\\right)$ |
> | $O\\left(\\frac{T}{C C\_{\\ell}} \\log \\frac{T}{C}\\right)$ | $O\\left(TN + \\left(\\frac{T}{C} C\_{\\ell} \\right) N^{2}\\right)$ |
>
> Here, $C\_{\\ell}$ denotes the number of levels processed simultaneously, that is, the amount of chunking along the level dimension.

---

### Meta-Review · Area_Chair_gR2f · 2026-01-11

**Summary:**

his paper introduces Log-Linear Attention, a framework that augments modern linear attention / SSM-style recurrent sequence models with a logarithmically growing set of hidden states instead of a single fixed-size recurrent state. Reviewers generally agree the paper is well-motivated and presents a coherent, intuitive idea: allocate more representational capacity near recent tokens while preserving longer-term memory at higher levels, thereby addressing the fixed-state bottleneck of linear RNN/SSM approaches. After carefully reading the paper, review and author responses, the AC agrees with the majority of the reviewers on accepting the paper.

**Reviewer Concerns:**

see Summary

**Reviewer Scores:**

see Summary

---

### Decision · Program_Chairs · 2026-01-26

Accept (Poster)